# A computational screen for alternative genetic codes in over 250,000 genomes

Yekaterina Shulgina[1], Sean R Eddy[1,2,3]*

[1]Department of Molecular and Cellular Biology, Harvard University, Cambridge, United States; [2]Howard Hughes Medical Institute, Harvard University, Cambridge, United States; [3]John A Paulson School of Engineering and Applied Sciences, Harvard University, Cambridge, United States

**Abstract** The genetic code has been proposed to be a 'frozen accident,' but the discovery of alternative genetic codes over the past four decades has shown that it can evolve to some degree. Since most examples were found anecdotally, it is difficult to draw general conclusions about the evolutionary trajectories of codon reassignment and why some codons are affected more frequently. To fill in the diversity of genetic codes, we developed Codetta, a computational method to predict the amino acid decoding of each codon from nucleotide sequence data. We surveyed the genetic code usage of over 250,000 bacterial and archaeal genome sequences in GenBank and discovered five new reassignments of arginine codons (AGG, CGA, and CGG), representing the first sense codon changes in bacteria. In a clade of uncultivated Bacilli, the reassignment of AGG to become the dominant methionine codon likely evolved by a change in the amino acid charging of an arginine tRNA. The reassignments of CGA and/or CGG were found in genomes with low GC content, an evolutionary force that likely helped drive these codons to low frequency and enable their reassignment.

## Editor's evaluation

This work is a substantial contribution to the important and fascinating field of genetic code diversification.

*For correspondence:
seaneddy@fas.harvard.edu

## Introduction

The genetic code defines how mRNA sequences are decoded into proteins. The ancient origin of the standard genetic code is reflected in its near-universal usage, once proposed to be a 'frozen accident' that is too integral to the translation of all proteins to change (*Crick, 1968*). However, the discovery of alternative genetic codes in over 30 different lineages of bacteria, eukaryotes, and mitochondria over the past four decades has made it clear that the genetic code is capable of evolving to some degree (*Knight et al., 2001a*; *Kollmar and Mühlhausen, 2017*).

The first alternative genetic codes were discovered by comparing newly sequenced genomes to amino acid sequences obtained by direct protein sequencing. Nonstandard codon translations were found this way in human mitochondria (*Barrell et al., 1979*), *Candida* yeasts (*Kawaguchi et al., 1989*), green algae (*Schneider et al., 1989*), and *Euplotes* ciliates (*Meyer et al., 1991*). Some reassignments of stop codons to amino acids were detected from DNA sequence alone, based on the appearance of in-frame stop codons in critical genes (*Yamao et al., 1985*; *Caron and Meyer, 1985*; *Cupples and Pearlman, 1986*; *Keeling and Doolittle, 1996*; *McCutcheon et al., 2009*; *Campbell et al., 2013*; *Záhonová et al., 2016*). As DNA sequence data have accumulated faster than direct protein sequences, computational methods have been developed to predict the genetic code from

**eLife digest** All life forms rely on a 'code' to translate their genetic information into proteins. This code relies on limited permutations of three nucleotides – the building blocks that form DNA and other types of genetic information. Each 'triplet' of nucleotides – or codon – encodes a specific amino acid, the basic component of proteins. Reading the sequence of codons in the right order will let the cell know which amino acid to assemble next on a growing protein. For instance, the codon CGG – formed of the nucleotides guanine (G) and cytosine (C) – codes for the amino acid arginine. From bacteria to humans, most life forms rely on the same genetic code. Yet certain organisms have evolved to use slightly different codes, where one or several codons have an altered meaning.

To better understand how alternative genetic codes have evolved, Shulgina and Eddy set out to find more organisms featuring these altered codons, creating a new software called Codetta that can analyze the genome of a microorganism and predict the genetic code it uses. Codetta was then used to sift through the genetic information of 250,000 microorganisms. This was made possible by the sequencing, in recent years, of the genomes of hundreds of thousands of bacteria and other microorganisms – including many never studied before.

These analyses revealed five groups of bacteria with alternative genetic codes, all of which had changes in the codons that code for arginine. Amongst these, four had genomes with a low proportion of guanine and cytosine nucleotides. This may have made some guanine and cytosine-rich arginine codons very rare in these organisms and, therefore, easier to be reassigned to encode another amino acid.

The work by Shulgina and Eddy demonstrates that Codetta is a new, useful tool that scientists can use to understand how genetic codes evolve. In addition, it can also help to ensure the accuracy of widely used protein databases, which assume which genetic code organisms use to predict protein sequences from their genomes.

DNA sequence. The core principle of most methods is to align genomic coding regions to homologous sequences in other organisms (creating multiple sequence alignments) and then to tally the most frequent amino acid aligned to each of the 64 codons. This approach led to the discovery of new genetic codes in screens of ciliates (*Swart et al., 2016*; *Heaphy et al., 2016*), yeasts (*Riley et al., 2016*; *Krassowski et al., 2018*), green algal mitochondria (*Noutahi et al., 2019*; *Žihala and Eliáš, 2019*), invertebrate mitochondria (*Telford et al., 2000*; *Abascal et al., 2006a*; *Li et al., 2019*), and stop codon reassignments in metagenomic data (*Ivanova et al., 2014*) and the development of software for specific phylogenetic groups (*Abascal et al., 2006b*; *Mühlhausen and Kollmar, 2014*; *Noutahi et al., 2017*). Some approaches, such as FACIL (*Dutilh et al., 2011*), have expanded phylogenetic breadth by using profile hidden Markov model (HMM) representations of conserved proteins from phylogenetically diverse databases such as Pfam (*El-Gebali et al., 2019*). However, a systematic survey of genetic code usage across the tree of life has not yet been possible. Existing methods are generally either (1) phylogenetically restricted to clades where multiple sequence alignments can be built for a predetermined set of proteins or (2) lacking sufficiently robust and objective statistical footing to enable a large-scale screen with high accuracy.

A potentially incomplete set of alternative genetic codes limits our ability to understand the evolutionary processes behind codon reassignment. One open question is why some codon reassignments reappear independently. Reassignment of the stop codons UAA and UAG to glutamine is the most common change in eukaryotic nuclear genomes, appearing at least five independent times (*Schneider et al., 1989*; *Keeling and Doolittle, 1996*; *Keeling and Leander, 2003*; *Karpov et al., 2013*; *Swart et al., 2016*). In bacteria, all of the known changes reassign the stop codon UGA to either glycine in the Absconditabacteria and Gracilibacteria (*Campbell et al., 2013*; *Rinke et al., 2013*) or tryptophan in the Mycoplasmatales, Entomoplasmatales (*Bové, 1993*), and several insect endosymbiotic bacteria (*McCutcheon et al., 2009*; *McCutcheon and Moran, 2010*; *Bennett and Moran, 2013*; *Salem et al., 2017*). These recurring changes may reflect constraints imposed by the existing translational machinery. The mechanism of codon reassignment may involve changes to tRNA anticodons or tRNA wobble nucleotide modifications (which together dictate anticodon-codon pairing), aminoacyl-tRNA synthetase recognition of cognate tRNAs, release factor binding of stop codons, among others, each

of which may bias which reassignments are easier to evolve. However, without a complete picture of genetic code diversity, it is hard to disentangle patterns of codon reassignment from observation bias. For instance, in-frame stop codons caused by a stop codon reassignment may be more easily detectable than a subtle change in amino acid conservation indicative of a sense codon reassignment.

Another open question is how a new codon meaning can evolve without disrupting the translation of most proteins. Reassigning a codon leads to the incorporation of the incorrect amino acid at all preexisting codon positions (*Crick, 1968*). Three evolutionary models differ in the pressure that drives substitutions to remove the codon from positions that cannot tolerate the new translation. In the 'codon capture' model, the codon is first driven to near extinction by pressures unrelated to reassignment, such as biased genomic GC content or genome reduction, which then minimizes the impact of reassignment on protein translation (*Osawa and Jukes, 1989*). This model was first proposed for the reassignment of the stop codon UGA to tryptophan in *Mycoplasma capricolum*, whose low genomic GC content (25% GC) in combination with small genome size (1 Mb) was thought to have driven the stop codon UGA to extremely low usage in favor of UAA and allowed 'capture' of UGA by a tryptophan tRNA (*Bové, 1993*; *Osawa and Jukes, 1989*). For larger nuclear genomes, other models have been proposed where codon usage changes occur concurrently with, and are driven by, changes in decoding capability. In the 'ambiguous intermediate' model, a codon is decoded stochastically as two different meanings in an intermediate step of codon reassignment, and this translational pressure induces codon substitutions at positions where ambiguity is deleterious (*Schultz and Yarus, 1994*; *Massey et al., 2003*). Extant examples of ambiguous translation support the plausibility of this model, such as yeasts that translate the codon CUG as both leucine and serine by stochastic tRNA charging (*Gomes et al., 2007*) or by competing tRNA species (*Mühlhausen et al., 2018*). Alternatively, the 'tRNA loss-driven reassignment' model proposes an intermediate stage where a codon cannot be translated efficiently, perhaps due to tRNA gene loss or mutation, creating pressure for synonymous substitutions specifically away from that codon, allowing it to be captured later by a different tRNA (*Mühlhausen et al., 2016*; *Sengupta and Higgs, 2005*). These three models are not mutually exclusive, and substitutions at the reassigned codon can occur due to a combination of these pressures.

Here, we describe Codetta, a computational method for predicting the genetic code that can scale to analyze thousands of genomes. We perform the first survey of genetic code usage in all bacterial and archaeal genomes, reidentifying all known codes in the dataset and discovering the first examples of sense codon changes in bacteria. All five reassignments affect arginine codons (AGG, CGA, and CGG) and provide clues to help us understand how alternative genetic codes evolve.

## Results

### Codetta: A computational method to infer the genetic code

We developed Codetta, a computational method that takes DNA or RNA sequences from a single organism and predicts an amino acid translation for each of the 64 codons. Codetta can analyze sequences from all domains of life, including bacteria, archaea, eukaryotes, organelles, and viruses, and the ability to confidently predict codon decodings depends on having protein-coding regions with recognizable homology. The general idea is to align the input nucleotide sequence to probabilistic profiles of conserved protein domains in order to obtain, for each of the 64 codons, a set of profile positions aligned to that codon. Each profile position has 20 probabilities describing the expected amino acid. For each of the 64 codons, we aggregate over the set of aligned profile positions to infer the single most likely amino acid decoding of the codon. Most previous approaches for genetic code prediction use the same basic idea (*Telford et al., 2000*; *Abascal et al., 2006b*; *Dutilh et al., 2011*; *Mühlhausen and Kollmar, 2014*; *Swart et al., 2016*; *Heaphy et al., 2016*; *Riley et al., 2016*; *Krassowski et al., 2018*; *Noutahi et al., 2019*), typically aligning the input sequence to multiple sequence alignments and using a simple rule to select the best amino acid for each codon.

With Codetta, we extend this idea to systematic high-throughput analysis by using a probabilistic modeling approach to infer codon decodings and by taking advantage of the large collection of probabilistic profiles of conserved protein domains (profile HMMs) in the Pfam database (*El-Gebali et al., 2019*). Profile HMMs are built from multiple sequence alignments, and the emission probabilities at each consensus column are estimates of the expected amino acid frequencies. The Pfam database contains over 17,000 profile HMMs of conserved protein domains from all three domains of life, which

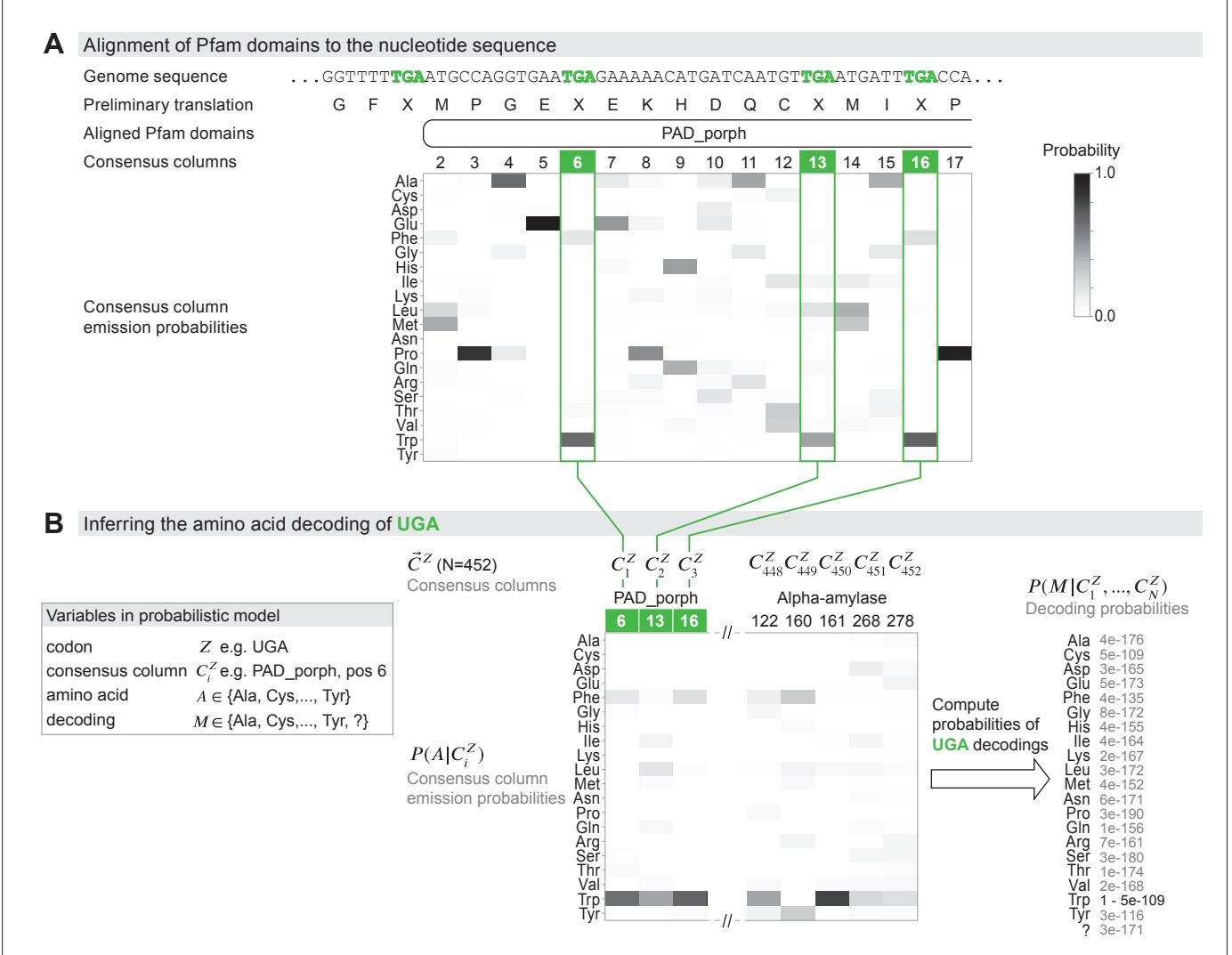

**Figure 1.** Schematic of the genetic code inference method implemented in Codetta. (**A**) A fragment of the *Mycoplasma capricolum* genome is used to demonstrate alignment of a Pfam domain (PAD_porph) to a preliminary standard code translation of the input DNA sequence (one of six frames shown). All canonical stop codons, including UGA (TGA in genome sequence, reassigned to tryptophan in *M. capricolum*), are translated as 'X' in the preliminary standard code translation that hmmscan (program used to align Pfam domains) treats as an unknown amino acid. Each consensus column in the PAD_porph domain has a characteristic emission probability for each of the 20 canonical amino acids, represented by a heatmap. (**B**) Pfam consensus columns aligning to UGA codons across the entire genome comprise the $\vec{C}^{Z}$ set for UGA ($N$ = 452 Pfam consensus columns). The Pfam emission probabilities $P(A|C_i^Z)$ for all 452 aligned consensus columns are used to compute the decoding probabilities $P(M|\vec{C}^Z)$. The most likely amino acid translation of UGA is inferred to be tryptophan, with decoding probability greater than the cutoff of 0.9999.

The online version of this article includes the following figure supplement(s) for figure 1:

**Figure supplement 1.** Estimates of error rate and power of Codetta inference.

are expected to align to about 50% of coding regions in a genome (*El-Gebali et al., 2019*). We align Pfam profile HMMs to a six-frame standard genetic code translation of the input DNA/RNA sequence using the HMMER hmmscan program (*Eddy, 2011*; *Figure 1A*). Since we rely on a preliminary standard code translation, conserved protein domains could fail to align in organisms using radically different genetic codes. In the set of statistically significant hmmscan alignments (E-value < $10^{-10}$), we make the simplifying approximation of considering each aligned consensus column independently, so the alignments are viewed as a set of pairwise associations between a codon $Z$ (64 possibilities) and a consensus column of a Pfam domain profile (denoted $C$, an index identifying a Pfam consensus column).

From these data, we infer each of the 64 codons one at a time (*Figure 1B*). For a codon $Z$ (e.g., UGA), the observed data $\vec{C}^Z$ are a set of $N$ consensus columns $C_i^Z$ ($i = 1...N$) that associate to $Z$ in the provisional alignments. We model the main data-generative process abstractly, imagining that each column $C_i^Z$ was drawn from the pool of all possible consensus columns by codon $Z$, which is translated as an unknown amino acid $A$. Each column has an affinity for codon $Z$ proportional to the column's emission probability for the amino acid $A$, $P(A|C)$. A consensus column strongly conserved for a particular amino acid $A$ will tend to only associate with codons that translate to $A$; moreover, consensus columns weakly conserved for $A$ may also associate with probability proportional to their conservation for $A$. Thus, this abstract-matching process generates an observed $C_i^Z$ column association with the codon $Z$ (translated as amino acid $A$) with probability

$$P(C_i^Z|A) = \frac{P(A|C_i^Z)P(C_i^Z)}{P(A)}.$$

Here, $P(A|C_i^Z)$ is the emission probability for amino acid $A$ at the Pfam consensus column $C_i^Z$. $P(A)$ is the average emission probability for amino acid $A$ over the pool of all possible consensus columns $C$, which we take to be all columns aligned to the target genome in order to better reflect genome-specific biases in amino acid usage.

Given the data $\vec{C}^Z$ and this abstract generative model, we infer the most likely decoding $M$ for codon $Z$ out of 21 possibilities $M \in \{\text{Ala}, \text{Cys}, ..., \text{Tyr}, ?\}$ (*Figure 1B*). The $M = ?$ model of nonspecific translation draws columns randomly and serves to catch codons that do not encode a specific amino acid, such as stop codons and ambiguously translated codons. For a given decoding $M$, the probability of the observed columns $\vec{C}^Z$ is then

$$P(\vec{C}^Z|M) = \begin{cases} \prod_{i=1}^N \frac{P(A=M|C_i^Z)P(C_i^Z)}{P(A=M)} & \text{if } M \in \{\text{Ala}, \text{Cys}, ..., \text{Tyr}\} \\ \prod_{i=1}^N P(C_i^Z) & \text{if } M = ? \end{cases}$$

Setting the prior probability of each decoding, $P(M)$, to be uniform, we compute the probability of the decoding $M$ as

$$P(M|\vec{C}^Z) = \frac{P(\vec{C}^Z|M)}{\sum_{M'} P(\vec{C}^Z|M')}$$

We assign an amino acid translation to a codon if it attains a decoding probability above some threshold (typically 0.9999). We assign a '?' if no amino acid decoding satisfies the probability threshold (including the case where '?' itself has high probability). A '?' assignment tends to happen if the codon is rare, with few aligned Pfam consensus columns on which to base the inference, or if the codon is ambiguously translated such that no single amino acid model reaches high probability. Because we do not model stop codons explicitly, we expect '?' to be the inferred meaning since stop codons ideally would have few or no aligned Pfam consensus columns.

To assess how many columns in $\vec{C}^Z$ are needed for reliable codon assignment, we constructed synthetic $\vec{C}^Z$ datasets ranging from 1 to 500 consensus columns by subsampling the Pfam consensus columns aligned to each of the 61 sense codons in the *Escherichia coli* genome. We calculated the per-codon error rate (fraction of samples predicting the incorrect amino acid) and the per-codon power (fraction of samples predicting the correct amino acid) using a probability threshold of 0.9999 (*Figure 1—figure supplement 1*). Lack of an amino acid inference ('?') was considered neither an error nor a correct prediction. Per-codon error rates were <0.00002 for all sizes of $\vec{C}^Z$. Depending on the codon, we found that about 8–34 aligned consensus columns suffice for >95% power to infer the correct amino acid. Accuracy may differ in real genomes for various biological reasons, but these results gave us confidence to proceed.

## Genetic code prediction of 462 yeast species confirms known distributions of CUG reassignment

We further validated Codetta on the budding yeasts (Saccharomycetes, 462 sequenced species) that vary in their translation of CUG as either serine, leucine, or alanine depending on the species (*Mühlhausen et al., 2016*; *Krassowski et al., 2018*; *Mühlhausen et al., 2018*). In some CUG-Ser clade

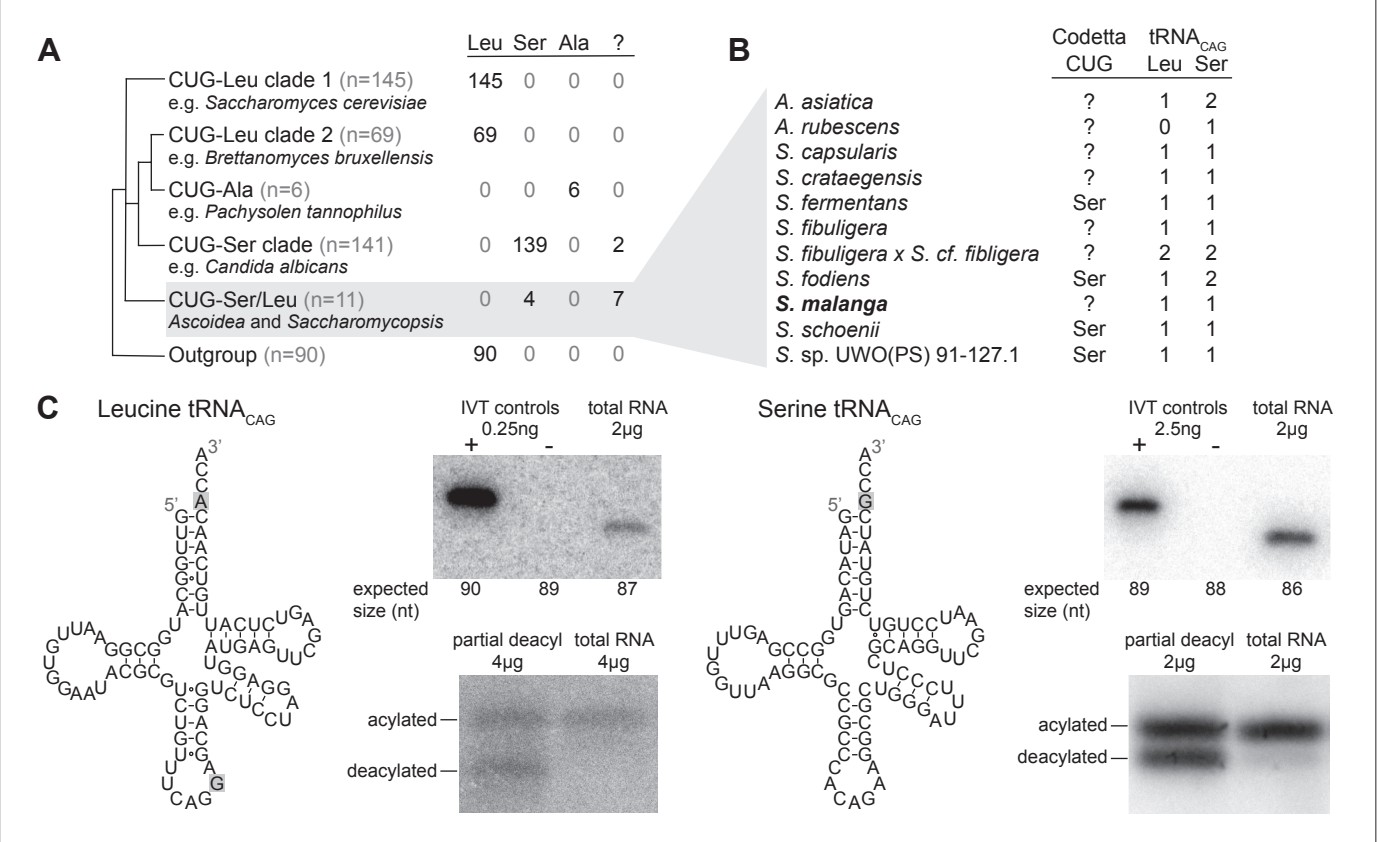

**Figure 2.** Analysis of CUG reassignments in yeast. (**A**) CUG translation inferred by Codetta of 462 Saccharomycetes species, grouped by phylogenetic clade. Cladogram was adapted from **Shen et al., 2018**. Phylogenetic placement of CUG-Leu/Ser clade is unresolved, thus the three-way branch. (**B**) Codetta CUG inference and number of tRNA_CAG genes in *Ascoidea* and *Saccharomycopsis* genomes. tRNA_CAG genes were identified using tRNAscan-SE 2.0 and were classified as being serine-type or leucine-type based on the presence of tRNA identity elements. (**C**) Northern blotting to confirm expression and charging of leucine and serine tRNA_CAG genes in *S. malanga*. Probable secondary structures of the two *S. malanga* tRNA_CAG are shown with features used for leucine/serine classification highlighted in gray. In the tRNA expression blots, in vitro transcribed (IVT) versions of the target tRNA (+ control) and the most similar other tRNA (– control, as determined by sequence homology with the probe) were used as controls for probe specificity. In the tRNA charging blots, a partial deacylation control was used to help visualize the expected band sizes for acylated and deacylated versions of the probed tRNA.

The online version of this article includes the following figure supplement(s) for figure 2:

**Source data 1.** Table of all analyzed yeast genomes with phylogenetic grouping and Codetta CUG inference.

**Source data 2.** Tarball of original blot images.

**Source data 3.** Table of in vitro transcription DNA template sequences.

**Figure supplement 1.** Distribution of Pfam consensus column amino acid support for CUG codons in *B. inositovora* and *C. fragrans*.

species, such as *Candida albicans*, CUG codons are stochastically decoded as a mix of serine (97%) and leucine (3%) because the CUG-decoding tRNA_CAG is aminoacylated by both the seryl- and leucyl-tRNA synthetases (**Suzuki et al., 1997**; **Gomes et al., 2007**). Codetta is not designed to predict ambiguous decoding and is expected to assign either the dominant amino acid or a '?' in cases like *C. albicans*.

For 453 species, the predicted CUG translation was consistent with the known phylogenetic distribution of CUG reassignments (**Figure 2A**). This includes *C. albicans*, which was predicted to use the predominant serine translation (**Gomes et al., 2007**). For the remaining nine species, Codetta did not put a high probability on any amino acid decoding of CUG (inferred meaning of '?'). Two of these species – *Babjeviella inositovora* and *Cephaloascus fragrans* – are basal members of the CUG-Ser clade. Both of these genomes contain a CUG-decoding tRNA_CAG gene with features of serine identity (see Materials and methods) and *B. inositovora* has previously been shown to translate CUG codons primarily as serine by whole proteome mass spectrometry (**Krassowski et al., 2018**; **Mühlhausen**

*et al., 2018*), suggesting that CUG is decoded as serine in these species. Codetta did not infer an amino acid for CUG because the aligned Pfam consensus columns were not consistently conserved for a single amino acid (*Figure 2—figure supplement 1*).

The other seven species without an inferred amino acid for CUG all belong to the closely related genera *Ascoidea* and *Saccharomycopsis* (four additional species in these clades were predicted to translate CUG as serine). Analysis of tRNA genes revealed that 10 out of 11 species in this clade encode two types of tRNA$_{CAG}$ genes, one predicted to be serine-type and one leucine-type, suggesting that CUG may be ambiguously translated as both serine and leucine via competing tRNAs in some of these species (*Figure 2B*). We used northern blotting to assay the expression of both tRNA$_{CAG}$ genes in some of these species under a variety of growth conditions (data not shown), but could detect reliable expression of both serine- and leucine-type tRNA$_{CAG}$ genes only in *Saccharomycopsis malanga* (only the serine tRNA$_{CAG}$ could be detected in other species) (*Figure 2C*). To determine whether both tRNAs are aminoacylated, we performed acid urea PAGE northern blotting that separates aminoacylated and deacylated tRNAs but does not identify the charged amino acid. We found that both serine and leucine *S. malanga* tRNA$_{CAG}$ are predominantly charged in cells (*Figure 2C*), likely partaking in the translation of CUG codons. If CUG is indeed translated ambiguously in this clade, it would explain why Codetta did not place a high probability on any single amino acid decoding for some species.

The existence of serine and leucine tRNA$_{CAG}$ genes in some *Ascoidea* and *Saccharomycopsis* yeasts was reported by *Krassowski et al., 2018* and *Mühlhausen et al., 2018* while we were conducting experiments. Ambiguous translation of CUG was demonstrated in *Ascoidea asiatica* (*Mühlhausen et al., 2018*); however, for *S. malanga* only expression of the serine tRNA$_{CAG}$ could be detected (*Krassowski et al., 2018*) and incorporation of predominantly serine at protein positions encoded by CUG (*Mühlhausen et al., 2018*). In contrast to these studies, we used a saturated growth condition where the leucine tRNA$_{CAG}$ seems to be more highly expressed. While we did not quantify the relative expression of the two tRNA$_{CAG}$ in *S. malanga*, a visual comparison of the band intensities in *Figure 2C* suggests that the expression of the leucine tRNA$_{CAG}$ is at least 10 times less than the serine tRNA$_{CAG}$ even in the saturated growth condition.

These results show that Codetta can correctly infer canonical and non-canonical codon translations and can flag unusual situations such as ambiguous translation even though it assumes unambiguous translation. All of the remaining 63 codons were inferred to use the expected translation in all species, with the following exceptions. In three species belonging to a lineage of *Hanseniaspora* with low genomic GC content (*Steenwyk et al., 2019*), the arginine codons CGC and/or CGG had a '?' inference due to few (<20) aligned Pfam consensus columns presumably due to rare usage of those codons. In eight other species, either the stop codon UAG or UGA was inferred to code for tryptophan due to

**Table 1.** A summary of all bacterial clades previously known to use a codon reassignment.

For each clade, the NCBI taxonomic IDs (taxids) shown most closely correspond to the known phylogenetic distribution from the literature. For each codon reassignment, we show the number of sequenced species analyzed by Codetta and how many were inferred to use the expected amino acid or had no inferred amino acid. None of the analyzed species belonging to reassigned clades were predicted to use an unexpected amino acid at the reassigned codon. [1] *Bové, 1993*, [2] *Volokhov et al., 2007*, [3] *McCutcheon et al., 2009*, [4] *Bennett and Moran, 2013*, [5] *McCutcheon and Moran, 2010*, [6] *Salem et al., 2017*, [7] *Rinke et al., 2013*, and [8] *Campbell et al., 2013*.

| Phylogenetic distribution | NCBI taxids | Reference | N species | Codon reassignment | Reassigned codon | |
|---|---|---|---|---|---|---|
| | | | | | Expected amino acid | Uninferred ('?') |
| Entomoplasmatales and Mycoplasmatales | 186328, 264638, 2085 | [1, 2] | 199 | UGA Stop→W | 191 | 8 |
| *Hodgkinia cicadicola* | 573658 | [3] | 1 | UGA Stop→W | 1 | 0 |
| *Nasuia deltocephalinicola* | 1160784 | [4] | 1 | UGA Stop→W | 1 | 0 |
| *Zinderia insecticola* | 884215 | [5] | 1 | UGA Stop→W | 1 | 0 |
| *Stammera capleta* | 2608262 | [6] | 1 | UGA Stop→W | 1 | 0 |
| Gracilibacteria | 363464 | [7] | 15 | UGA Stop→G | 13 | 2 |
| Absconditabacteria | 221235 | [8] | 6 | UGA Stop→G | 6 | 0 |

**Table 2.** A summary of codon inferences from the bacterial and archaeal genomes analyzed by Codetta, dereplicated to one assembly per species.

The Codetta inference for each codon is compared against a genetic code annotation derived by layering the known bacterial genetic codes in *Table 1* over the NCBI taxonomy. Reassigned stop codons are included with sense codons. Values can be calculated from *Supplementary file 1*.

| | | Bacteria | | Archaea | |
|---|---|---|---|---|---|
| | | 46,384 species | | 2309 species | |
| Sense | Total (N codons × N species) | 2,829,648 | | 140,849 | |
| | Expected amino acid | 2,823,497 | 99.78% | 140,631 | 99.85% |
| | Other amino acid | 612 | 0.02% | 0 | 0.00% |
| | Uninferred ('?') | 5539 | 0.20% | 218 | 0.15% |
| Stop | Total (N codons × N species) | 138,928 | | 6927 | |
| | Amino acid | 290 | 0.21% | 9 | 0.13% |
| | Uninferred ('?') | 138,638 | 99.79% | 6918 | 99.87% |

some (<23) aligned Pfam consensus columns. We could not find any nuclear suppressor tRNA genes, and we believe that these inferences are due to the erroneous alignment of Pfam domains to in-frame stop codons in pseudogenes. In-frame stop codons do not appear randomly within pseudogenes but instead are most likely to result from single-nucleotide transitions from certain codons (such as the UGG tryptophan codon).

## Computational screen of bacterial and archaeal genomes finds previously known alternative genetic codes

To explore the diversity of genetic codes in bacterial and archaeal genomes, we used Codetta to analyze 251,571 assembled genomes from GenBank, including partial assemblies and those derived from single-cell genomics and metagenomic assembly. Summaries of our analysis (*Table 1* and *Table 2*) are shown for a subset of the results, dereplicated to reduce the over-representation of frequently sequenced organisms by selecting a single assembly for each species-level NCBI taxonomic ID (48,693 unique species: 46,384 bacteria, 2309 archaea). Results for the full dataset and the dereplicated subset are available in *Supplementary file 1*.

To see if our screen recovered known alternative genetic codes, we collated a comprehensive literature summary of all bacterial and archaeal clades known to use alternative genetic codes (*Table 1*) and layered it over the NCBI taxonomy, annotating all remaining organisms with the standard genetic code. This resulted in a genetic code annotation for each species. For most species using known alternative genetic codes in our dataset, our predictions at the reassigned codon agreed with the expected amino acid translation (*Table 1*). There were no instances of reassigned codons predicted to translate as an unexpected amino acid, but there were a few cases of reassigned UGA codons that had no amino acid meaning inferred ('?' inference).

Since the uninferred codons could represent a lack of sensitivity by Codetta, we looked more closely at these examples. In the Mycoplasmatales and Entomoplasmatales, which are believed to translate the canonical stop codon UGA as tryptophan (*Bové, 1993*), eight species had no inferred amino acid meaning for UGA due to fewer than four aligned Pfam consensus columns. All of these genomes lack a UGA-decoding $tRNA_{UCA}^{Trp}$ gene and all but one instead contain a release factor 2 gene (which terminates translation at UGA). Five of these species are included in the Genome Taxonomy Database (GTDB) (*Parks et al., 2020*), a comprehensive phylogeny of over 190,000 bacterial and archaeal genomes, where they are grouped into a different order (GTDB order RF39). We therefore attribute at least five (and perhaps all eight) as a taxonomic misannotation in the NCBI database, and we believe that UGA is a stop codon in these species. In the Gracilibacteria, which are believed to translate the stop codon UGA as glycine (*Rinke et al., 2013*), two species had no inferred amino acid meaning for UGA. Neither genome contained the expected UGA-decoding $tRNA_{UCA}^{Gly}$ gene and both instead encoded a release factor 2 gene, supporting that UGA

is a stop codon and not a glycine codon in these species. Indeed, one of these species is included in the GTDB and is grouped in a different order than the other UGA-reassigned Gracilibacteria and Absconditabacteria.

Across the 48,693 genomes (dereplicated to one assembly per species), we predicted the amino acid translation of a total of 2,970,497 individual sense codons (roughly 61 times the number of genomes), with 99.79% of the predictions consistent with the expected amino acid (similar proportion across bacteria and archaea) (*Table 2*). About 0.19% of sense codons had a '?' inference, demonstrating that entire genomes contain more than enough information to infer the amino acid translation of most sense codons. Unexpected amino acid meanings were predicted for 612 sense codons. These are candidates for new codon reassignments, but could also include inference errors. For stop codons, 99.80% out of a total of 145,855 stop codons across the dereplicated bacterial and archaeal genomes had no inferred amino acid meaning, as expected. 290 bacterial stop codons and 9 archaeal stop codons were inferred to translate as an amino acid, adding to our list of candidate new genetic codes.

## Validation of candidate new alternative genetic codes

To prioritize high-confidence novel genetic codes, we gathered additional evidence by examining (1) the translational components (tRNA and/or release factor genes) involved in the reassignment, (2) the usage of the reassigned codon, including manual examination of alignments of highly conserved single-copy genes, and (3) the phylogenetic extent of the proposed reassignment. Since many candidate genetic codes were found in uncultivated clades with only rough taxonomic classification on NCBI, we explored phylogenetic relationships using the GTDB (*Parks et al., 2020*). The GTDB is a phylogeny of over 190,000 archaeal and bacterial genomes, providing provisional domain-to-species phylogenetic classifications for uncultivated as well as established clades. A list of all candidate novel genetic codes can be found in *Supplementary file 2*. We focused on the candidate codon reassignments with the highest degree of additional evidence and attempted to characterize common sources of error. The set of lower-confidence candidates may still include additional real codon reassignments requiring further validation.

The most common error was the inference of AGA and/or AGG arginine codons as coding for lysine, occurring in 567 bacterial species. Almost all of the AGA- and AGG-decoding tRNAs found in these genomes were consistent with arginine identity (based on the arginine identity elements A/G73 and A20), supporting that AGA and AGG are arginine codons in the majority of these species. The unusually high GC content of these genomes (ranging between 0.52 and 0.77, median 0.68) suggests that the source of the lysine inference may come from high GC content-driven nonsynonymous substitutions of the AAA and AAG lysine codons to AGA and AGG arginine codons at protein residues that can tolerate either positively charged amino acid. As a result, AGA and AGG codons would consistently appear at residues conserved for lysine in other species, which Codetta would mistake for the signature of codon reassignment. Genomic GC content has long been correlated with greater arginine usage and lower lysine usage (*Sueoka, 1961*; *Lightfield et al., 2011*), possibly due to substitutions between the aforementioned lysine and arginine codons (*Knight et al., 2001b*). This error could be mitigated in future analyses by using profile HMMs built from sequences that match the analyzed genome in GC content or amino acid composition.

Some erroneous stop codon inferences resulted from genome contamination by organisms with known stop codon reassignments. We suspected contamination when the Pfam consensus columns aligned to a stop codon were only present in a limited part of the genome and confirmed the origin of these regions by homology search of the genes containing the in-frame stop codons. We have found examples of predicted stop reassignments in *Sulfolobus* assemblies caused by contamination with UGA-recoding *Mycoplasma* contigs, in an alphaproteobacteria assembly caused by contamination with UAA- and UAG-recoding ciliate contigs, in *Chloroflexi* assemblies caused by contamination with UGA-recoding Absconditabacteria contigs, and in others.

We found five clades using candidate novel alternative genetic codes with additional computational evidence, such as tRNA genes consistent with the new translation. All five new genetic codes involve the reassignment of arginine codons, representing the first sense codon reassignments in bacteria.

## Reassignment of the canonical arginine codon AGG to methionine in a clade of uncultivated Bacilli

Eight bacterial genomes were inferred to translate AGG, a canonical arginine codon, as methionine. All eight genomes were assembled from fecal metagenomes of baboons or humans (*Parks et al., 2017*; *Almeida et al., 2019*) and have only coarse-grained NCBI genome classification as uncultured Bacillales or Mollicutes bacteria. The GTDB assigns these eight genomes to a three species clade within the placeholder genus UBA7642 (family CAG-288, order RFN20, class Bacilli), of which all other species were inferred to translate AGG as arginine (*Figure 3A*).

In each of the reassigned genomes, the AGG inference by Codetta is based on a sufficiently large number of aligned Pfam consensus columns (over 2200 compared to an average of about 1800 for each of the other 60 sense codons) from over 480 different Pfam domains. *Figure 3B* shows an example multiple sequence alignment of uridylate kinase, a single-copy conserved bacterial gene, from the reassigned species, outgroup genomes, and several more distantly related bacteria. In the alignment, AGG codons are used interchangeably with AUG methionine codons within the reassigned clade and tend to occur at positions conserved for methionine and other nonpolar amino acids in the other species.

In the reassigned clade, AGG is the dominant methionine codon with a usage of 209–235 per 10,000 codons in Pfam alignments, outnumbering the canonical methionine codon AUG (59–69 per 10,000 codons) (*Figure 3A*). The process of codon reassignment involves genome-wide codon substitutions to remove the reassigned codon from positions that cannot tolerate the new amino acid, leading to depressed codon usage. High usage of AGG in the reassigned clade suggests that this is an established codon reassignment that has had time to rebound in frequency through synonymous substitutions with the standard AUG methionine codon. In many outgroup genomes, AGG is a rare arginine codon (*Figure 3A*).

Escape from viral infection has been put forth as a potential selective pressure for the evolution of alternative genetic codes, although viruses are also known to infect some alternative genetic code organisms such as *Mycoplasma* and mitochondria (*Shackelton and Holmes, 2008*). We inferred the genetic code of phage genomes assembled by *Al-Shayeb et al., 2020* from the same baboon fecal metagenomic dataset as some reassigned Bacilli genomes. Two phage assemblies were predicted to translate AGG as methionine (assemblies GCA_902730795.1 and GCA_902730815.1). The assemblies do not contain genes for the AGG-decoding tRNA$_{CCU}$, so the phage presumably rely on the host tRNAs for translation. Thus, some phage may have adapted to the AGG translation as methionine in the reassigned Bacilli.

We used tRNAscan-SE 2.0 (*Chan et al., 2021*) to determine which tRNAs are available to decode AGG in the reassigned and outgroup genomes (*Figure 3A*). Some tRNA genes are missing, possibly due to the incomplete nature of some metagenome-assembled genomes as indicated by low genome completeness estimates. The cognate tRNA for the AGG codon, tRNA$_{CCU}$, from the reassigned clade has features of methionine identity (including an A73 discriminator base and G2:C71 and C3:G70 base pairs in the acceptor stem) and lacks the important arginine identity element A20 in the D-loop (*Meinnel et al., 1993*; *Giegé et al., 1998*), supporting translation of AGG as methionine (*Figure 3C*). In vitro experiments have shown that anticodon mutations to tRNA$_{CAU}^{Met}$ disrupt recognition by the *E. coli* methionyl-tRNA synthetase (MetRS); however, the C35 change necessary to decode the AGG codon affects the least critical anticodon nucleotide (*Schulman and Pelka, 1983*). To see if any compensatory changes have occurred in the MetRS from the reassigned clade to accommodate recognition of a new anticodon, we compared the predicted MetRS sequence to that of *Aquifex aeolicus* (crystal structure in complex with tRNA$_{CAU}$, PDB 2CSX/2CT8, *Nakanishi et al., 2005*). The three residues that contact the anticodon nucleotides in *A. aeolicus*, which are conserved through all domains of life (*Nakanishi et al., 2005*), also remain unchanged in the reassigned clade MetRS sequence (*Figure 3—figure supplement 1*).

The outgroup genomes contain a tRNA$_{CCU}$ with features of arginine identity (including a G73 discriminator base and A20 in the D-loop). The genomic context of the tRNA$_{CCU}$ is similar in many reassigned clade and outgroup genomes, flanked by a tRNA$_{CCG}^{Arg}$ immediately downstream and a homolog of GenBank protein CDA36808.1 upstream (*Figure 3D*). This implies that the reassigned and outgroup tRNA$_{CCU}$ evolved from the same ancestral tRNA gene, and the reassigned methionine tRNA$_{CCU}$ likely emerged through a change in aminoacylation of an arginine tRNA$_{CCU}$ rather than through duplication and anticodon mutation of a methionine tRNA.

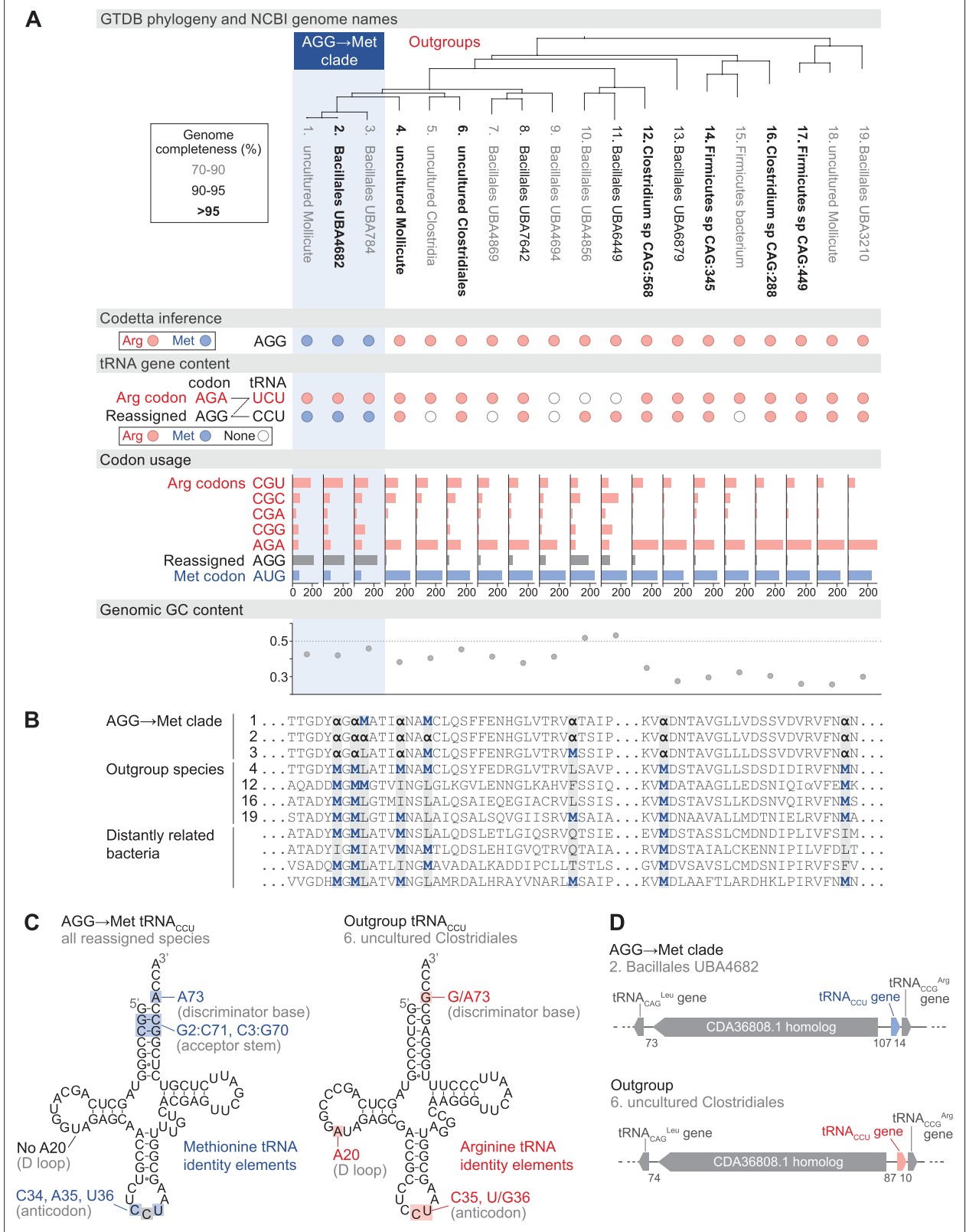

**Figure 3.** Reassignment of AGG from arginine to methionine in a clade of uncultivated Bacilli. (**A**) Genome Taxonomy Database (GTDB) phylogenetic tree of the Bacilli AGG→Met clade and closest outgroup genomes, with the annotated NCBI genome name shaded according to the GTDB CheckM estimated genome completeness. GTDB genus UBA7642 corresponds to species #1–9, and GTDB family CAG-288 corresponds to species #1–16. For each genome, the Codetta AGG inference is indicated by colored circles (red: arginine; blue: methionine). The presence of tRNA genes is also indicated

*Figure 3 continued*

by filled circles for tRNA$_{UCU}$ and tRNA$_{CCU}$, colored by the predicted amino acid charging based on known identity elements (see Materials and methods), or a white circle if no tRNA gene could be detected. The lines connecting codons and anticodons represent the likely decoding capabilities, with dashed lines representing likely weaker interactions. Codon usage is the frequency per 10,000 codons aligned to Pfam domains. (**B**) Multiple sequence alignment of uridylate kinase (BUSCO POG091H02JZ) from the reassigned species, selected outgroup species, and four more distantly related bacteria (*Bacillus subtilis*, *Nostoc punctiforme*, *Chlamydia caviae*, and *Escherichia coli*). All AGGs are represented by α. Alignment regions containing multiple nearby AGG positions in the reassigned species are shown. (**C**) A comparison of the AGG-decoding tRNA$_{CCU}$ in the Bacilli AGG→Met clade (identical sequence in all genomes) and in an outgroup genome (#6, uncultured Clostridiales). tRNA sequence features involved in methionine identity in the reassigned clade tRNA$_{CCU}$ and arginine identity in the outgroup tRNA are highlighted (***Meinnel et al., 1993***; ***Giegé et al., 1998***), with nucleotide numbering following the convention of ***Sprinzl et al., 1998***. The C35 anticodon nucleotide in the AGG→Met clade tRNA is highlighted in gray because it does not match the A35 methionine identity element. (**D**) The genomic context surrounding the tRNA$_{CCU}$ gene in a member of the Bacilli AGG→Met clade (#2, Bacillales UBA4682) and in an outgroup species (#6, uncultured Clostridiales). Gene lengths and intergenic distances are drawn proportionally, with the number of base pairs between each gene indicated below.

The online version of this article includes the following figure supplement(s) for figure 3:

**Source data 1.** Table of genome accessions, Codetta AGG inference, tRNA gene presence, codon usage, and genome GC content for the reassigned AGG→Met Bacilli and outgroup species.

**Figure supplement 1.** Alignment of MetRS sequences.

The reassigned genomes use an arginine-type tRNA$_{UCU}$ to decode the unaffected AGA arginine codon. Depending on the post-transcriptional modification of the U34 anticodon nucleotide, the arginine tRNA$_{UCU}$ could recognize AGG via wobble and potentially cause ambiguous translation. In *E. coli*, the U34 of tRNA$_{UCU}$ is modified to 5-methylaminomethyluridine (***Sakamoto et al., 1993***), which primarily decodes the AGA codon with a low level of AGG recognition (***Spanjaard et al., 1990***). ***Mukai et al., 2015*** demonstrated that it is possible to engineer separate decodings for AGA and AGG in *E. coli* by reducing expression level of the tRNA$_{UCU}$ to the point where decoding of AGG by tRNA$_{UCU}$ is presumably insignificant in competition with the cognate tRNA$_{CCU}$. In most outgroup genomes, AGA is the dominant arginine codon, while in the reassigned clade the preferred arginine codon is CGU (***Figure 3A***), which may indicate reduced demand and expression of tRNA$_{UCU}$ to avoid ambiguous translation of AGG. A similar potential for ambiguous translation due to U34 wobble exists with the previously known decoding of UGA as glycine and UGG as tryptophan in Absconditabacteria and Gracilibacteria (***Campbell et al., 2013***; ***Rinke et al., 2013***).

## Reassignments of arginine codons CGA and CGG occur in clades with low genomic GC content

The remaining four clades with codon reassignments supported by additional computational evidence all affect the arginine codons CGA and/or CGG (***Figure 4***). Three clades are in the phylum Firmicutes: the genus *Peptacetobacter* is predicted to translate CGG as glutamine (***Figure 4—figure supplement 1***), a clade of uncultivated Bacilli in the GTDB order RFN20 (same as the AGG-reassigned Bacilli) is predicted to translate CGG as tryptophan (***Figure 4—figure supplement 2***), and members of the genus *Anaerococcus* are also predicted to translate CGG as tryptophan (***Figure 4—figure supplement 3***). The fourth clade is Absconditabacteria (also known as Candidate Division SR1, part of the Candidate Phyla Radiation), which is predicted to have reassigned CGA and CGG both to tryptophan (***Figure 4—figure supplement 4***), in addition to the already known reassignment of UGA from stop to glycine.

In contrast to the reassignment of AGG to become the dominant methionine codon (described in the previous section), these CGA/CGG reassignments resemble earlier stages of codon reassignment where the reassigned codon has not yet rebounded in frequency through synonymous substitutions with the new amino acid. Due to the rarity of the reassigned CGA/CGG codons, these predictions are based on fewer aligned Pfam consensus columns and may be more prone to error. As a check for each reassignment, we looked for examples of the reassigned codon in conserved regions of single-copy gene alignments (***Figure 4—figure supplement 1B***, ***Figure 4—figure supplement 2B***, ***Figure 4—figure supplement 3B*** and ***Figure 4—figure supplement 4B***) and found multiple supporting positions for all reassigned codons except the extremely rare CGG codon in *Anaerococcus*. We also looked for tRNA genes with an anticodon and amino acid identity elements consistent with the reassignment (***Figure 4—figure supplement 1***, ***Figure 4—figure supplement 2***, ***Figure 4—figure supplement 3***

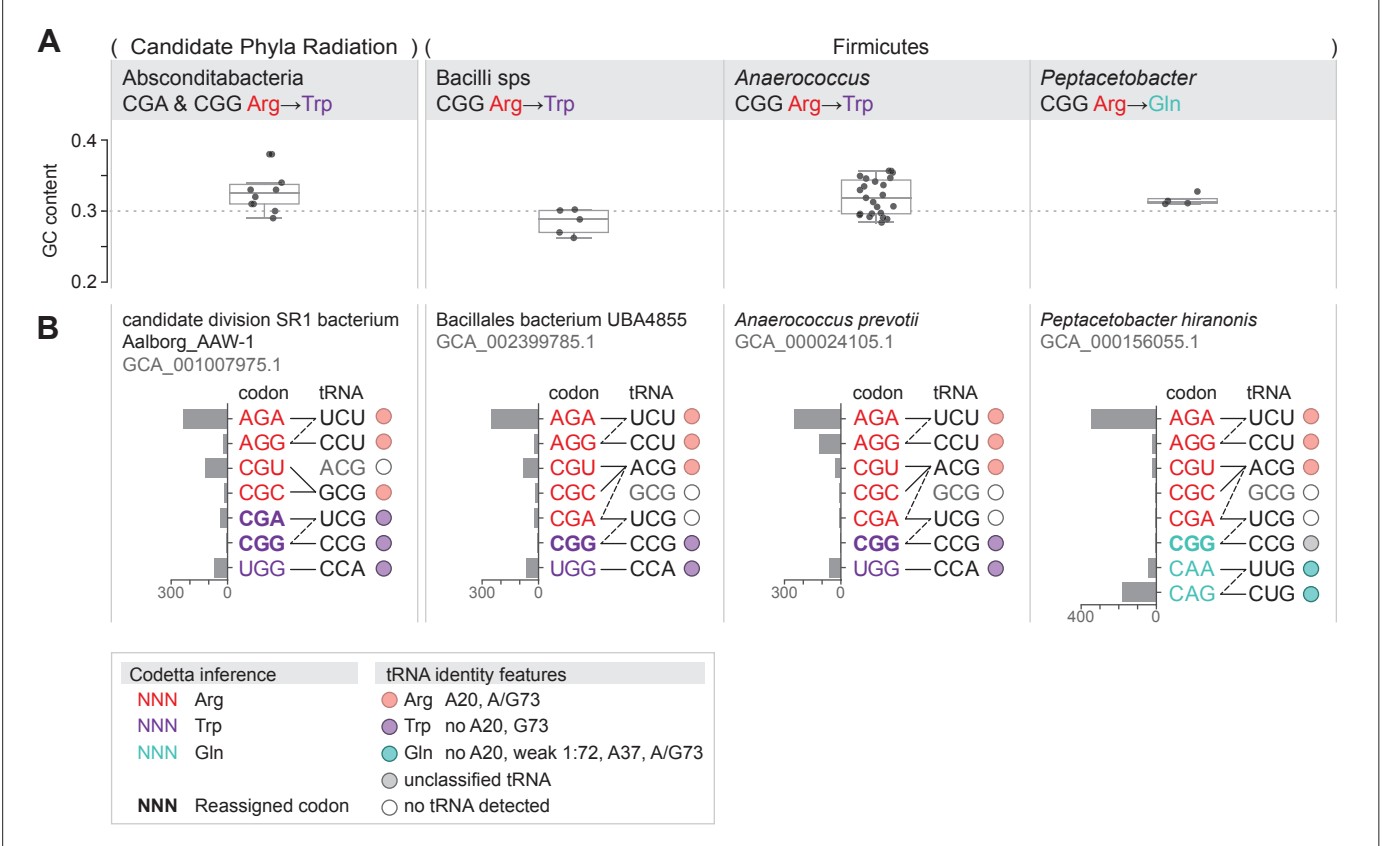

**Figure 4.** Summary of GC content, codon usage, and tRNA genes of four CGA and/or CGG reassignments. (**A**) Distribution of genomic GC contents across all species in the reassigned clades. (**B**) For each reassigned clade, we selected a representative species to show codon usage and tRNA decoding ability. Codon usage is plotted for the reassigned codon and for all other codons of the original and new amino acids in usage per 10,000 in Pfam alignments. Codons are colored by their Codetta inference and reassigned codons are bolded. The lines connecting codons and tRNA anticodons represent the likely decoding capabilities, with dashed lines representing likely weaker interactions. The anticodon ACG is presumed to be modified to ICG, and UCG is presumed to be modified in a way that restricts wobble to CGA and CGG, but could potentially recognize CGU and CGC as well depending on the true modification state. Anticodons in gray font are not expected to be found in the respective clade. Presence of tRNA genes is indicated by filled circles, colored by the predicted amino acid charging based on the identity elements in the key.

The online version of this article includes the following figure supplement(s) for figure 4:

**Source data 1.** Table for each CGA/CGG reassignment containing genome accessions, Codetta CGA/CGG inference, tRNA gene presence, codon usage, and genome GC content for the reassigned clade and outgroup species.

**Figure supplement 1.** Reassignment of CGG→Gln in *Peptacetobacter*.

**Figure supplement 2.** Reassignment of CGG→Trp in a clade of Bacilli.

**Figure supplement 3.** Reassignment of CGG→Trp in *Anaerococcus*.

**Figure supplement 4.** Reassignment of CGA and CGG→Trp in Absconditabacteria.

and *Figure 4—figure supplement 4*), and found consistent tRNAs for all clades except for *Peptacetobacter* whose CGG-decoding tRNA$_{CCG}$ resembles neither an arginine or glutamine isotype. While amino acid conservation at the reassigned codon and sequence-based prediction of tRNA charging may lend support to a predicted codon reassignment, only experimental confirmation can establish how the reassigned codons are translated in vivo and whether there is ambiguous translation. In particular, *Anaerococcus* and *Peptacetobacter* include culturable species and may be experimentally confirmed in the future.

The four CGA/CGG candidate reassignments share several features that suggest common evolutionary forces at play. Most notable is the very low genomic GC content of the reassigned clades (0.26–0.38, *Figure 4A*). In all four clades, the usage of GC-rich CGN-box codons – including CGA and CGG – is depressed and arginine residues are primarily encoded by AGA codons (*Figure 4B*).

In the three Firmicute CGG reassignments, CGG is an extremely rare codon (codon usage <6 per 10,000 in aligned Pfam domains for all species). In the Absconditabacteria, CGG also tends to be quite rare (<7 per 10,000 in all but one species) with CGA slightly more abundant (<37 per 10,000 in all species). In one Absconditabacteria (assembly GCA_002791215.1), the frequency of both CGA and CGG approaches the frequency of the canonical tryptophan codon UGG, consistent with a more advanced stage of codon reassignment (usage of CGA and CGG is 30 and 24 per 10,000, compared to 35 for UGG). Low genomic GC content is thought to be created by mutational bias in favor of AT nucleotides, causing a gradual shift towards synonymous codons with lower GC compositions (*Knight et al., 2001b*; *Muto and Osawa, 1987*). This may have helped disfavor usage of CGA and/or CGG prior to reassignment, lessening the impact of changing the codon meaning.

The tRNAs used to decode the CGN codon box may have also influenced the reassignment of CGA and CGG codons. A shared feature of the three Firmicute CGG reassignments is that the tRNA$_{UCG}$ is missing (*Figure 4B*), presumably lost prior to or during the reassignment of CGG. If the tRNA$_{UCG}$ were present, it would likely recognize both CGA and CGG via wobble that would complicate assigning different amino acid meanings to those two codons. In the absence of tRNA$_{UCG}$, CGA (along with CGU and CGC) is presumably decoded by a tRNA$_{ICG}^{Arg}$ (derived by deamination of tRNA$_{ACG}^{Arg}$, the only widespread instance of inosine tRNA wobble in bacteria). This leaves CGG to be decoded solely by a tRNA$_{CCG}$ (*Figure 4B*). In this situation, CGG is one of a few codons in the genetic code decoded by a single dedicated tRNA, potentially facilitating codon reassignment since the translational meaning of CGG can now be altered independently of neighboring codons. The inosine wobble modification is not used by some deeply branching bacteria (*Rafels-Ybern et al., 2019*), and the tRNA$_{ACG}^{Arg}$ gene appears to be lacking in the Candidate Phyla Radiation, including Absconditabacteria. Instead, these bacteria use a tRNA$_{GCG}^{Arg}$ to decode CGU and CGC, and rely on a tRNA$_{UCG}$ and tRNA$_{CCG}$ to recognize CGA and CGG (*Figure 4B*). Since the ability of tRNA$_{UCG}$ to decode CGA and CGG makes it difficult to split the translational meanings of the two codons, it may explain why both CGA and CGG are reassigned to tryptophan together in the Absconditabacteria.

For some of these reassignments, close outgroup species may shed light on potential intermediate stages of codon reassignment. The CGG reassignment in the Absconditabacteria may extend to members of the sister clade Gracilibacteria – some Gracilibacteria were predicted to translate CGG as tryptophan, while others translate CGG as arginine (*Figure 4—figure supplement 4*). This may reflect a complicated history of CGG reassignment and possible reversion to arginine translation. For the CGG reassignment in *Peptacetobacter*, the closest sister group (which includes the pathogen *Clostridioides difficile*) has extremely rare usage of CGG (<1 per 10,000 in aligned Pfam domains in all but two species) and appears to lack any tRNA capable of decoding CGG by standard codon-anticodon pairing rules (*Figure 4—figure supplement 1*). This may resemble an intermediate stage in codon reassignment before the ability to translate CGG as a new amino acid is gained, similar to the unassigned CGG codon in *M. capricolum* (*Oba et al., 1991*). In *Anaerococcus*, all species contain a CGG-decoding tRNA$_{CCG}$ with features of tryptophan identity (*Figure 4—figure supplement 3*). Unexpectedly, members of an outgroup genus *Finegoldia* also have a tRNA$_{CCG}$ with features of tryptophan identity (CGG inferred to be '?' by Codetta). It is unclear if the tRNA$_{CCG}$ genes in these two clades share an evolutionary history or represent independent events.

## Discussion

We present a method for computationally inferring the genetic code that can scale to analyze hundreds of thousands of genomes that we call Codetta. We validate Codetta on the well-studied reassignments of CUG in yeasts and rediscover the likely ambiguous translation of CUG as serine and leucine in some *Ascoidea* and *Saccharomycopsis* species. We conduct the first systematic survey of genetic code usage across the majority of sequenced organisms, analyzing all sequenced bacteria and archaea (over 250,000 assemblies). The five new alternative genetic codes described here substantially expand the known diversity of codon reassignments in bacteria. Now, in addition to reassignments of the stop codon UGA to tryptophan or glycine, we have the first sense codon reassignments in bacteria, affecting the arginine codons AGG, CGA, and CGG. Two reassignments occur in culturable bacteria – in *Anaerococcus* and *Peptacetobacter* – and could be experimentally confirmed in the future, for example by proteomic mass spectrometry.

Since Codetta selects the most likely amino acid translation among the 20 canonical amino acids, some types of codon reassignments may be missed. We cannot predict reassignment to a noncanonical amino acid – for such codons, Codetta would pick the nonspecific model or an amino acid that is used similarly in other species. We also cannot directly detect ambiguous translation, which may represent an important stage in codon reassignment. However, the failure to infer an amino acid translation despite a significant number of aligned Pfam consensus columns may hint at ambiguous translation, as was the case for CUG in *Ascoidea* and *Saccharomycopsis*. Since we do not model translational initiation and termination, we cannot detect the use of new start and stop codons or context-dependent stop codons that also possess an amino acid meaning, known to occur in some eukaryotes (*Swart et al., 2016*; *Heaphy et al., 2016*; *Záhonová et al., 2016*).

Expanding our analysis to eukaryotic, organellar, and viral genomes will help fill in the diversity of alternative genetic codes, but poses additional challenges. Since we align profile HMMs to a six-frame translation of the entire genome, the pervasive pseudogenes in many eukaryotic genomes will likely increase the rate of incorrect codon inferences by having sufficient homology for alignment but enough accumulated mutations to cause incorrect pairing of codons to consensus columns. Smaller scale surveys of eukaryotic genetic code diversity have focused on transcriptomic datasets (*Swart et al., 2016*; *Heaphy et al., 2016*), which may alleviate this problem. Some viral and organellar genomes have very few protein-coding genes that may limit the ability to confidently infer the entire genetic code. One strategy is to improve sensitivity at the cost of generalizability by using clade-specific profile HMMs instead of Pfam, which may increase the proportion of aligned coding sequence. Another challenge in some organellar genomes is extensive mRNA editing (*Gray, 1996*; *Alfonzo et al., 1997*), which violates our assumption that the genomic codon sequence represents the mRNA sequence and may require analyzing the edited transcriptome to ensure correct correspondence of codons to profile HMM positions.

In the 'codon capture' model of codon reassignment, genome-wide pressures such as biased GC content or genome reduction drive a codon to near extinction such that the codon can acquire a new tRNA decoding with a minimal effect on translation (*Osawa and Jukes, 1989*). Most UGA reassignments in bacteria occur in clades with very low genomic GC content, which is thought to have reduced UGA to very low usage in favor of the stop codon UAA. This includes the Mycoplasmatales and Entomoplamatales (0.24–0.39 GC) (*Jukes, 1985*; *Bové, 1993*), Absconditabacteria and Gracilibacteria (0.21–0.53 GC, *Figure 4—source data 1*; *Campbell et al., 2013*; *Rinke et al., 2013*), and most insect endosymbiotic bacterial reassignments (0.13–0.17 GC, except for *Hodgkinia cicadicola* with 0.28–0.58 GC) (*McCutcheon and Moran, 2010*; *Bennett and Moran, 2013*; *Salem et al., 2017*; *Campbell et al., 2015*). The CGA and/or CGG reassignments described here similarly exhibit low genomic GC content (0.26–0.38) and very rare usage of GC-rich codons including CGA and CGG. A codon does not need to completely disappear for reassignment to be facilitated by rare codon usage, and it is likely that a brief period of translational ambiguity or inefficiency helps drive the remaining codon substitutions. We posit that, in bacteria, reduction in codon usage driven by genome-wide processes plays a major role in enabling codon reassignment and may explain why codon reassignments repeatedly evolve in clades such as Firmicutes (known for their low genomic GC content) and lifestyles such as endosymbiosis (which is often accompanied by genome reduction and low GC content) (*McCutcheon and Moran, 2011*).

All five of the new reassignments affect arginine codons (AGG, CGA, and CGG). While these are the first instances of arginine codon reassignment in non-organellar genomes, several arginine reassignments are known in mitochondria: in various metazoan mitochondria, the codons AGA and AGG have been reassigned to serine, glycine, and possibly stop and AGG has been reassigned to lysine (*Knight et al., 2001a*; *Abascal et al., 2006a*), and in various green algal mitochondria AGG has been reassigned to alanine and methionine and CGG to leucine (*Noutahi et al., 2019*). Arginine codons have several unique features that may predispose them to codon reassignment. First, across the tree of life, arginine has an over-representation of codons in the genetic code relative to usage in proteins (*Jukes et al., 1975*; *King and Jukes, 1969*), contributing to rare usage of the least favored arginine codon. Second, the six arginine codons range from 1 to 3 GC nucleotides in composition (only equaled by leucine), which may create greater bias in codon usage in response to genomic GC content than for amino acids with less GC variability in their codons. In organisms with small genomes, these features alone might make the rarest arginine codon very low in number and more susceptible

than other codons to reassignment. The arginine codon CGG may be even more of a target for reassignment because, in most bacteria, the only widespread instance of inosine tRNA wobble is used to decode the CGU, CGC, and CGA arginine codons (*Grosjean et al., 2010*). In the absence of a $tRNA_{UCG}$, CGG is decoded by a dedicated $tRNA_{CCG}$ and can be reassigned without affecting the translation of other codons.

Some codon reassignments have convergently reappeared across the tree of life: CGG to tryptophan in three bacterial clades described here, AGG to methionine in a clade of Bacilli described here and in green algal mitochondria (*Noutahi et al., 2019*), UGA to tryptophan in multiple bacterial, mitochondrial, and eukaryotic lineages (*Knight et al., 2001a*), and others. Recurrent changes could reflect (1) a common evolutionary process, for example, low GC content-driven reassignments disproportionately affecting codons sensitive to GC fluctuations, or (2) shared constraints imposed by conserved translational machinery, including tRNAs and aminoacyl-tRNA synthetases. For example, the tRNA anticodon-codon pairing rules dictate that U- and C-ending codons cannot be assigned separate meanings, and indeed this has not been observed in any known genetic codes. This may explain why in low GC content genomes we see reassignments of the arginine codon CGG but not the arginine codon CGC, which would have to be reassigned together with CGU. The selection of amino acid changes in the codon reassignments described here is not clearly explained by biochemical similarity (except possibly for the reassignment of CGG from arginine to glutamine). The amino acid choice may be related to the constraints on evolving new tRNA anticodons. Most of the changes described here (and indeed all of the changes known in bacteria) involve a single nucleotide difference from cognate anticodons: $tRNA_{CCU}$ in addition to $tRNA_{CAU}^{Met}$ for the AGG to methionine reassignment, $tRNA_{CCG}$ in addition to $tRNA_{CCA}^{Trp}$ for the CGG to tryptophan reassignments, and $tRNA_{CCG}$ in addition to $tRNA_{CUG}^{Gln}$ for the CGG to glutamine reassignment. Evolving a new anticodon through a single mutation may be more probable than through multiple mutations. However, the methionine $tRNA_{CCU}$ involved in the reassignment of AGG in a clade of Bacilli appears to have evolved from an arginine $tRNA_{CCU}$ through mutations that altered aminoacyl-tRNA synthetase recognition, rather than by an anticodon mutation to a methionine $tRNA_{CAU}$ gene. Alternatively, this pattern could result from a limitation on the new anticodons that an aminoacyl-tRNA synthetase could accept since most aminoacyl-tRNA synthetases use the anticodon in part to distinguish cognate and non-cognate tRNAs (*Giegé et al., 1998*). Upon characterizing the diversity of genetic codes in other parts of the tree of life, we may discover that the general patterns and evolutionary pressures differ from bacteria, reflecting differences in translational machinery, lifestyle, or genome characteristics.

## Materials and methods
### Computational inference of the genetic code from nucleotide sequence
A preliminary translation of the input nucleotide sequences is produced by first breaking any long sequences into nonoverlapping 100 kb pieces (because of a limit on input protein sequence length for hmmscan), then translating into all six frames (as six polypeptide sequences) using the standard genetic code with stop codons translated as 'X.' A custom version of Pfam 32.0 profiles was produced from Pfam seed alignments using hmmbuild ``--enone``, which turns off entropy weighting, resulting in emission probability parameters closer to the original amino acid frequencies in the input alignments. Significant homologous alignments were identified by searching each translated polypeptide against the custom Pfam database using hmmscan from HMMER 3.1b2 (*Eddy, 2011*) for domain hits with E-value $< 10^{-10}$.

Alignments were further filtered to remove uncertainly aligned consensus columns (posterior probabilities of alignment < 95%). By default, no single Pfam consensus columns was allowed to account for more than 1% of total aligned consensus columns for a codon in order to mitigate some artifacts such as repetitive pseudogene families in some genomes; when this happened, the number of codon positions aligned to that specific consensus column was downsampled to 1% of the total (if a codon was aligned to fewer than 100 Pfam consensus columns total, then each unique consensus columns was downsampled to one occurrence). We excluded hits to five classes of Pfam models including mitochondrial proteins, viral proteins, selenoproteins, pyrrolysine-containing proteins, and proteins belonging to transposons and other mobile genetic elements. These filtered sets of aligned consensus columns defined the input $\vec{C}$ sets for each codon. The equations from the main text are

then used (in log-probability calculations for numerical stability) to infer $P(M|\vec{C}^Z)$ for each codon, with a default decoding probability threshold of 0.9999.

The computational requirements are dominated by the hmmscan step, which takes about an hour on a single CPU core for an ~12 Maa six-frame translation of a typical 6 Mb bacterial genome. We ran different genomes in parallel on a 30,000 core computing resource, the Harvard Cannon cluster. We implemented this method as Codetta v1.0, a Python 3 program that can be found at https://github.com/kshulgina/codetta/releases/tag/v1.0, (copy archived at swh:1:rev:4f5f31a33beed19b-c3e10745154705ad002273df, *Yekaterina, 2021*).

## Measuring error rate and power on synthetic datasets

A six-frame translation of the *E. coli* O157:H7 str. Sakai genome (GCA_000008865.2) was searched against the custom --enone Pfam 32.0 profile database as described above. We generated 1000 random subsamples each of 1 through 50, 100, and 500 aligned consensus columns per sense codon and inferred the most likely decoding as described above. A codon inference was considered 'true' (T) if the correct amino acid meaning was inferred, 'false' (F) if an incorrect amino acid meaning was inferred, and 'uninferred' (U) if the nonspecific decoding was most probable or if no model surpassed the model probability threshold. For a given model probability threshold, per-codon error rate is the fraction of samples with a false inference (F/(T + F +  U)). Per-codon power is the fraction of samples with a true inference (T/(T + F +  U)). Both values were evaluated individually for each sense codon and also aggregated across all sense codons.

## Genetic code inference of archaeal and bacterial genomes

Assembly identifiers for all archaeal and bacterial genomes were downloaded from the NCBI Genome database on June 4, 2020, and Codetta analysis was performed on all archaeal and bacterial genome assemblies. Genetic code inference results for all analyzed genomes can be found in *Supplementary file 1*. A variety of additional files supporting new genetic codes are available at https://github.com/kshulgina/ShulginaEddy_21_genetic_codes, (*Shulgina, 2021*, copy archived at swh:1:rev:a2bf2a1ef0bcd5ea0319354ab6e9cbba89f9e934).

We used the NCBI taxonomy database (downloaded on July 15, 2020) to cross-reference all assemblies with taxonomic identifiers. All analyzed genome assemblies from GenBank are associated with an NCBI taxonomic ID (taxid). Because some of these taxids correspond to subspecies or strain-level designations, we assigned a species-level taxid to each assembly by iteratively stepping up the NCBI taxonomy until a species-level node was reached. To create a dereplicated dataset, we picked one genome assembly per NCBI species-level taxid. If multiple genome assemblies were associated with an NCBI species-level taxid, assemblies were sorted based on RefSeq category (reference, representative, or neither) and then genome completeness level and a single-genome assembly was randomly selected from the highest ranked category.

Phage assemblies derived from the same metagenomic samples as the AGG-recoding Bacilli were obtained by identifying the phage assemblies from *Al-Shayeb et al., 2020* whose sample accessions were linked to the metagenomic sequencing experiments SRX834619, SRX834622, SRX834629, SRX834636, SRX834653, SRX834655, or SRX834666. Codetta analysis of the phage genomes was performed as described above.

## Cross-referencing the NCBI taxonomy with known distributions of genetic code usage

A complete list of bacterial clades previously known to use alternative genetic codes was collated with corresponding references for genetic code discovery and taxonomic distribution (*Table 1*). For each clade, we determined a set of NCBI taxids best defining the phylogenetic extent of each reassigned clade. We used this to generate a curated genetic code annotation for all NCBI species-level taxids: for the taxids defining each reassigned clade, all species-level child nodes were annotated with the alternative genetic code; all remaining species-level taxids were annotated with the standard genetic code.

We used the GTDB (version R05-RS95; *Parks et al., 2020*) to determine the phylogenetic placement of species that use candidate new genetic codes and to identify the most closely related outgroup species.

## Identification of tRNA genes and other translational components

The tRNA gene content of genomes was determined by running tRNAscan-SE 2.0 (*Chan et al., 2021*) with default settings and a tRNA model appropriate for the domain of life (i.e., option -E for eukaryotes, -B for bacteria, -A for archaea). To help ensure that tRNAs of interest were not missed, we also ran a low-stringency search with the general tRNA model and no cutoff score (options -G -X 0) and a search with previous version of tRNAscan-SE (option -L) and manually examined the outputs.

We searched bacterial genomes for release factor genes using the TIGRFAM 15.0 (*Haft et al., 2013*) release factor 2 model (TIGR00020) and release factor 1 model (TIGR00019) and for the methionyl-tRNA synthetase gene using the TIGRFAM 15.0 model (TIGR00399) with hmmscan against a six-frame translation of the entire genome with default settings. Since the release factors are homologs, if the two models hit overlapping genomic coordinates, we kept the hit with the more significant E-value. Methionyl-tRNA synthetase alignments were generated using MAFFT v7.429 (*Katoh and Standley, 2013*) with default settings.

For the AGG arginine to methionine reassignment in a clade of Bacilli, we classified $tRNA_{CCU}$ genes as being primarily arginine acceptors if the tRNA had A20 in the D-loop and a A/G73 discriminator base, and primarily methionine acceptors if the tRNA had an A73 discriminator base and not A20 in the D-loop (*Giegé et al., 1998*). The weaker methionine identity elements G2:C70, C3:G69 in the acceptor stem were used to support the assignment (*Meinnel et al., 1993*). In the reassignment of CGG to glutamine in *Peptacetobacter*, we classified tRNAs as arginine-type using the rules above, and as glutamine-type if the tRNA was missing arginine identity element A20 and contained the set of glutamine identity elements consisting of a weak 1:72 basepair, A37, and A/G73 (*Jahn et al., 1991*). We took the additional glutamine identity elements G2:C71 and G3:C70 in the acceptor stem, G38 in the anticodon loop, and G10 in the D-stem as support for glutamine identity (*Jahn et al., 1991*; *Hayase et al., 1992*). For the reassignments of CGA and/or CGG arginine to tryptophan, we classified tRNAs as primarily arginine acceptors using the rules above, and provisionally as tryptophan acceptors if the tRNA had a G73 discriminator base and not A20 in the D-loop (*Giegé et al., 1998*). We considered the weak tryptophan identity element A/G1:U72 in the acceptor stem as support for tryptophan identity but did not require it (*Himeno et al., 1991*). In the Absconditabacteria and Gracilibacteria, we classified $tRNA_{UCA}$ genes as glycine acceptors if the tRNA had G1:C72, C2:G71, G3:C70 in the acceptor stem and U73 discriminator base (*Giegé et al., 1998*). We refrained from assigning identity if the tRNA did not fit the above patterns or if the D-loop sequence was unusual such that it was unclear which nucleotide is N20. D-loop and variable loop insertions were placed at positions following the convention of *Sprinzl et al., 1998*.

## Multiple sequence alignment of BUSCO genes

For some candidate novel alternative genetic codes, we constructed multiple sequence alignments of conserved single-copy bacterial genes from the BUSCO database v3 (*Waterhouse et al., 2018*). To identify orthologs of a BUSCO gene in a particular genome, we first created a dataset of putative protein sequences by translating all open reading frames longer than 50 codons using the inferred genetic code (assuming standard stop codons unless reassigned), with candidate reassigned codons translated as 'X.' Then, we queried each of the 148 bacterial BUSCO profile HMMs against all putative proteins using hmmsearch from HMMER 3.1b2 with default settings and an E-value cutoff of $10^{-13}$, and picked the most significant hit if it also yielded a reciprocal best hit against the entire BUSCO profile HMM database using hmmscan with the same E-value cutoff. Multiple sequence alignments were generated using MAFFT v7.429 (*Katoh and Standley, 2013*) with default settings.

For the described novel genetic codes, BUSCO alignments containing the reassigned codon in the reassigned clade were individually inspected and alignments containing the reassigned codon at conserved positions in well-aligned regions were preferentially selected as example alignments.

## Annotation of genomic context

To determine the genomic context surrounding the $tRNA_{CCU}$ gene in the uncultivated Bacilli predicted to have reassigned AGG to methionine and in close outgroup genomes, we predicted tRNA and protein coding genes in the whole genome as described above. We annotated each putative protein coding gene with the reciprocal best hit homolog among annotated protein-coding genes in the outgroup assembly GCA_000434395.1 using phmmer from HMMER 3.1b2 with a $10^{-10}$ E-value cutoff.

## Phylogenetic grouping and Codetta analysis of CUG usage by budding yeasts

For analysis of CUG translation in budding yeasts, we selected all genomes belonging to the class Saccharomycetes (NCBI taxid 4891), which represent 463 unique NCBI species taxids with at least one genome. The genomes were dereplicated to one assembly per species-level taxid as described above. Yeast species were split into six taxonomic categories based on the 'major clade' annotation from the phylogenetic analysis by *Shen et al., 2018* as follows: outgroups (major clades: Lipomyceta-ceae, Trigonopsidaceae, Dipodascaceae/Trichomonascaceae, Alloascoideaceae, Sporopachydermia), CUG-Leu clade 1 (major clades: Phaffomycetaceae, Saccharomycodaceae, Saccharomycetaceae), CUG-Leu clade 2 (major clade: Pichiaceae), CUG-Ser (major clade: CUG-Ser1), CUG-Ala (major clade: CUG-Ala), and CUG-Ser/Leu (major clade: CUG-Ser2). Species that were not included in the analysis by *Shen et al., 2018* were sorted into the same major clade as other members of their annotated genus on NCBI. A single species (*Candida* sp. JCM 15000) could not be placed into a category and was excluded from the analysis. The expected CUG translation for each clade follows *Shen et al., 2018* and is consistent with other studies of CUG translation (*Riley et al., 2016*; *Krassowski et al., 2018*; *Mühlhausen et al., 2018*). Genetic codes were predicted by Codetta as described above. A table describing all yeast genomes analyzed can be found in *Figure 2—source data 1*.

## Identification of tRNA genes and isotype classification in yeasts

tRNA gene content of yeast genomes was determined using tRNAscan-SE 2.0 as described above. In eukaryotes, only leucine- and serine-tRNAs have a long (>12 nucleotide) variable loop so we used this feature to confirm the $tRNA_{CAG}$ identity as serine or leucine. In yeast, serine tRNAs typically have a conserved G73 discriminator base but can tolerate any nucleotide (*Himeno et al., 1997*), while leucine tRNA identity is conferred by a A73 discriminator base and A35 and G37 in anticodon loop (*Soma et al., 1996*). We categorized $tRNA_{CAG}$ genes as either serine-acceptors or leucine-acceptors based on the presence of these features. In some CUG-Ser clade species, serine CAG-tRNAs containing a G37 have been found to be charged with leucine at a low level (3%) (*Suzuki et al., 1997*); for categorization purposes, we would consider these tRNAs to be primarily serine-acceptors.

## *S. malanga* growth and RNA extraction

*S. malanga* (NRRL Y-7175) was obtained from the Agricultural Research Service Culture Collection (Peoria, IL). Cells were inoculated into 5 mL of YPD liquid media (containing 1% yeast extract, 2% peptone, and 2% dextrose) from a colony on a YPD agar plate and grown to saturation for 4 days at 25°C on rotating wheel.

Total RNA was extracted in acidic conditions to preserve tRNA charging, following the steps outlined in *Varshney et al., 1991* with the following modifications. Cells were harvested by centrifugation (5 min at 4000 rpm at 4°C), resuspended in 500 μL ice-cold buffer containing 0.3 M NaOAc pH 4.5 and 10 mM EDTA and added to 500 μL ice-cold phenol:chloroform (pH 4.5) and 500 μL of 0.4–0.5 μm acid-washed glass beads for cell lysis. All RNA extraction steps were performed at 4°C. In the first round of extraction, cells were vortexed for 30 min, rested on ice for 3 min, centrifuged for 15 min at 20,000×g, and the aqueous layer was transferred to 500 μL of phenol:chloroform (pH 4.5), which was subject to a second round of extraction (identical, except for 3 min vortex). A last round of extraction was performed in 500 μL of chloroform with a 15 s vortex and 2 min centrifugation. RNA in the aqueous phase was precipitated and resuspended in buffer containing 10 mM NaOAc pH 4.5 and 1 mM EDTA.

## Northern blotting for tRNA expression

The single-stranded DNA probes used for detection of *S. malanga* $tRNA_{CAG}^{Ser}$ (5′ GAAATCCCAGCGCCTTCTGTGGGCGGCGCCTTAACCAAACTCGGC 3′) and *S. malanga* $tRNA_{CAG}^{Leu}$ (5′ TTGACAATGAGACTCGAACTCATACCTCCTAG 3′) were 5′ end-labeled with [$\gamma$-P$^{32}$]-ATP by T4 polynucleotide kinase (New England Biosciences) and purified using ProbeQuant G-50 Micro Columns (GE Healthcare Life Sciences).

In vitro transcribed tRNAs were used as controls for probe specificity. For the *S. malanga* $tRNA_{CAG}^{Ser}$ probe, an in vitro transcribed version of the target $tRNA_{CAG}^{Ser}$ was used as a positive control and an in vitro transcribed version of $tRNA_{CGU}^{Ser}$ was used as a control for cross-hybridization. For the $tRNA_{CAG}^{Leu}$

probe, an in vitro transcribed version of the target tRNA $^{\text{Leu}}_{\text{CAG}}$ was used as a positive control and an in vitro transcribed version of tRNA $^{\text{Leu}}_{\text{CAA}}$ was used as a control for cross-hybridization. Cross-hybridization controls were selected by aligning the reverse complement of the probe sequence using MAFFT v7.429 (*Katoh and Standley, 2013*) with default settings to all tRNA genes in the *S. malanga* genome (found by tRNAscan-SE 2.0) and selecting the non-target tRNA with the highest pairwise alignment score. In vitro transcribed tRNAs were produced using the MAXIscript T7 Transcription Kit (Thermo) from a DNA template composed of a T7 promoter (5′ GATCTAATACGACTCACTATAGGGAGA 3′) followed by the tRNA sequence (*Figure 2—source data 3*). The resulting tRNA transcript has an additional six nucleotides of the promoter included at the 5′ end. CCA-tails were not included in the in vitro transcribed tRNA sequences.

Total RNA and in vitro transcribed controls for probe specificity were denatured in formamide buffer (Gel Loading Buffer II, Thermo) at 90°C for 5 min and electrophoretically separated on a 10% TBE urea gel (Novex). Gels were rinsed in 0.5× TBE and RNA was transferred onto a Hybond N + membrane (GE Healthcare) in 0.5× TBE by semi-dry transfer (Bio-Rad Transblot) at 3 mA/cm² for 1 hr. Blots were crosslinked on each side using a Stratalinker UV crosslinker on the 'auto-crosslink' setting. Blots were prehybridized in PerfectHyb Plus Hybridization buffer (Sigma) at 64°C for 1 hr prior to incubation with the radiolabeled DNA probe overnight. Blots were washed at 64°C twice in low-stringency buffer (0.1% SDS, 2× SSC) for 15 min and once in high-stringency buffer (0.1% SDS, 0.1× SSC) for 10 min, exposed on storage phosphor screens, and scanned using a Typhoon imager.

## Acid urea PAGE northern blotting for tRNA charging

For the partial deacylation control, total RNA was treated in 100 mM Tris pH 7.0 at 37°C for 30 min, quenched with an equal volume of buffer containing 50 mM NaOAc and 100 mM NaCl, and precipitated. Electrophoresis on acid urea polyacrylamide gels was performed as described in *Varshney et al., 1991*. 4 µg of total RNA and partial deacylation control in acid urea sample buffer (0.1 NaOAc pH 4.5, 8 M urea, 0.05% bromophenol blue, 0.05% xylene cyanol) were loaded onto a 0.4 mm thick 6.5% polyacrylamide gel (SequaGel) containing 8 M urea and 100 mM NaOAc pH 4.5 and run for 18 hr at 450 V in 4°C with 100 mM NaOAc pH 4.5 running buffer. The region between the two dyes corresponds to the tRNA size range, and was cut out and transferred onto a blot for probing following the same steps as above for northern blotting.

## Acknowledgements

We thank members of the Eddy lab for discussions and for comments on the manuscript, A Murray and A Darnell for advice and guidance on northern blotting experiments, and R Helmiss for feedback on data presentation. Computations were performed on the Cannon cluster, supported by the Harvard FAS Division of Science's Research Computing Group.

## Additional information

### Funding

| Funder | Grant reference number | Author |
| --- | --- | --- |
| National Human Genome Research Institute | F31-HG010984 | Yekaterina Shulgina |
| National Human Genome Research Institute | R01-HG009116 | Sean R Eddy |
| Howard Hughes Medical Institute | | Sean R Eddy |

The funders had no role in study design, data collection and interpretation, or the decision to submit the work for publication.

## Author contributions
Yekaterina Shulgina, Conceptualization, Data curation, Formal analysis, Funding acquisition, Investigation, Methodology, Software, Validation, Visualization, Writing - original draft, Writing - review and editing; Sean R Eddy, Conceptualization, Funding acquisition, Methodology, Project administration, Supervision, Writing - review and editing

## Author ORCIDs
Yekaterina Shulgina (iD) http://orcid.org/0000-0001-7658-9294
Sean R Eddy (iD) http://orcid.org/0000-0001-6676-4706

## Decision letter and Author response
Decision letter https://doi.org/10.7554/eLife.71402.sa1
Author response https://doi.org/10.7554/eLife.71402.sa2

## Additional files

### Supplementary files
• Supplementary file 1. Table of predicted genetic codes by Codetta across all analyzed genome assemblies.

• Supplementary file 2. Table of all candidate novel genetic codes predicted by Codetta across all analyzed genome assemblies.

• Transparent reporting form

### Data availability
Results of computational analyses performed in this study are included in the manuscript and supporting files. Source data files have been provided for Figures 2, 3, and 4.

The following previously published datasets were used:

| Author(s) | Year | Dataset title | Dataset URL | Database and Identifier |
|---|---|---|---|---|
| Sayers EW, Cavanaugh M, Clark K, Ostell J, Pruitt K, Karsch-Mizrachi I | 2020 | GenBank | https://www.ncbi.nlm.nih.gov/assembly/ | NCBI Assembly, NCBI |

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
