## [Editor Report]

This work is a substantial contribution to the important and fascinating field of genetic code diversification.

---

## [Decision Letter]

**Decision letter after peer review:**

Thank you for submitting your article "A computational screen for alternative genetic codes in over 250,000 genomes" for consideration by *eLife*. Your article has been reviewed by 2 peer reviewers, and the evaluation has been overseen by a Reviewing Editor and Molly Przeworski as the Senior Editor. The reviewers have opted to remain anonymous.

The reviewers are unanimous in their view that the proposed method for inference of alternative genetic codes is valid and will be useful for many researchers involved in analysis of genomic and metagenomic data. As such, this is a strong paper that requires revisions primarily for clarification.

Essential revisions:

1) Clarify the methodology and, in particular, address the following question of Reviewer 2: "How many occurrences of an apparently wrong amino acid are needed in practice to draw an inference?"

2) Clarify the situation with the mitochondrial codes as requested by Reviewer 2.

*Reviewer #1 (Recommendations for the authors):*

As I described in my comments, I strongly recommend the authors to validate amino acid assignment of CUG codon in S. malanga.

Other Comments:

Line 160: Explain the meaning of the sentence, "Lack of an amino acid inference ("?") contributed to neither.", in more detail.

Line 213-216 and Line 241: It is better to explain why the number of the aligned Pfam consensus columns was so small in these species.

Line 221: In the title of this paragraph, "Computational screen of all bacterial and archaeal genomes finds previously known alternative genetic codes", the authors used "all", but it is not precise. It should be removed.

Line 238-240: Provide the reference(s) for the claim that Mycoplasmatales and Entomoplasmatales translates the opal stop codon UGA as Trp.

Line 247-250: Provide the claim that Gracilibacteria translate UGA as Gly.

Line 282-284: Provide the reference for the argument that "high GC content-driven nonsynonymous substitutions of the AAA and AAG lysine codons to AGA and AGG arginine codons at protein residues that can tolerate either positively-charged amino acid".

Line 287: Citing N. Sueoka (1961) that "…have long been observed to preferentially use more arginine and less lysine.". However, in his original study, Sueoka is simply plotting the possible correlation between the GC contents and the amino acid composition, drawing from a limited 'dataset' at his time. He did not directly observe any preferential usage of Arg over Lys in high-GC bacteria. This sentence should be corrected accordingly, given the citation to Sueoka's paper.

Line 300-303: The authors should be more specific in their analysis of the values of probabiltiy and tRNA sequences on which they base the results they are describing here.

Line 317-318: The values of the probability on which the author's decision were based should be indicated or summarized.

Line 368-376: The values of the probability on which the author's decision were based should be indicated or summarized.

Line 408-414: Do these species in which tRNAUCG is missing have homologs of the modifying enzyme responsible for the formation of inosine?

Line 443: The authors described "two differently charged tRNAs" here. However, they did not demonstrate that.

Figure 2 and line 709-715 (method section): Supply the exact sequences of T7 in vitro transcribed tRNA used as controls in northern blot.

*Reviewer #2 (Recommendations for the authors):*

The paper proposes a method for large scale genome analysis that is able to detect reassignments in the genetic code. In general, the examples given are convncing. The method detects several variants that are already known and discovers some new ones. This appears to be useful development and a thorough analysis.

The essentials of the method are in lines 112-126. I don't find this 100% clear. If I understand correctly, the DNA sequence of the test sequence is translated with the standard genetic code and then aligned with HMMs of proteins. If the codon follows the standard code, the amino acid will be in agreement with the common amino acid in the alignment column. If the codon has been reassigned, the apparent amino acid will be an unusual one according to the alignment profile. This requires the protein sequence to be less variable than the genetic code. I can see that this would become convincing if the same apparently wrong amino acid appears consistently in alignment columns where a particular amino acid is the most common one. How many occurrences of an apparently wrong amino acid are needed in practice to draw an inference? This paragraph says there are 17000 alignments in Pfam, but it does not say how many are used for the inference.

There is no mention of mitochondrial genetic codes. Does this method work with mitochondrial genomes? Maybe there are too few coding sequences in mitochondria?

The case of Lys and Arg codons discussed in lines 277-291 is interesting. Since Lys and Arg are similar amino acids we expect non-synonymous substitutions to be frequent between Lys and Arg. Therefore it is not surprising that the method might give a false positive prediction of a codon reassignment. In fact it is surprising that more cases like this do not occur. For example AUA is frequently reassigned from lle to Met in mitochondria, and an Ile to Met non-synonymous substitution is also quite possible. Does this not also show up in the analysis?

At the end of the day, computational predictions of codon reassignments will not be fully certain until there is experimental confirmation. But this method seems very useful at suggesting cases that are worth looking at experimentally.

---

## [Author Response]

Essential revisions:1) Clarify the methodology and, in particular, address the following question of Reviewer 2: "How many occurrences of an apparently wrong amino acid are needed in practice to draw an inference?"

About 8 to 34 occurrences are enough for 95% power, depending on the codon. To better address this comment, we expanded upon our study of this question using subsampled data in a paragraph in the results (pg.4, lines 159-168). Briefly, we inferred the most likely amino acid decoding from different sized sets of aligned Pfam consensus columns, created by subsampling Pfam consensus columns that have aligned to codons in the *Escherichia coli* genome. Codetta treats all possible decodings for a codon a priori as equally likely, so it makes no difference whether the amino acid is canonical or “apparently wrong”. Our results showed that across sets of Pfam columns drawn from codons for different amino acids and sets of different sizes (ranging from 1 to 500 Pfam consensus columns), having at least 8 to 34 aligned consensus columns resulted in a correct inference >95% of the time. We added a figure supplement to Figure 1 showing the proportion of correct amino acid inferences as a function of the number of aligned consensus columns for each of the 61 sense codons. We updated the results text (pg.4, lines 159-168) and methods text (pg.18, lines 579-589) accordingly.

2) Clarify the situation with the mitochondrial codes as requested by Reviewer 2.

Codetta can predict the genetic code from any nucleotide sequence input, which includes mitochondrial genomes. The power to infer the decoding of a codon depends on the number of times the codon occurs in coding regions. An example of a mitochondrial genome analyzed by Codetta is in the Codetta usage instructions on GitHub, which walk through the process of analyzing the mitochondrial genome of the green algae *Pycnococcus provasolii*. 55 out of 62 sense codons are correctly inferred and the remaining 7 sense codons are left uninferred presumably due to the low number of aligned Pfam columns.

To clarify that Codetta can be applied to any nucleotide sequence input, and not just the bacterial, archaeal, and yeast genomes analyzed in this study, we added this text to the description of Codetta:

“Codetta can analyze sequences from all domains of life, including bacteria, archaea, eukaryotes, organelles, and viruses, and the ability to confidently predict codon decodings depends on having protein-coding regions with recognizable homology.” (pg. 3, lines 104-107)

Reviewer #1 (Recommendations for the authors):As I described in my comments, I strongly recommend the authors to validate amino acid assignment of CUG codon in S. malanga.Other Comments:Line 160: Explain the meaning of the sentence, "Lack of an amino acid inference ("?") contributed to neither.", in more detail.

We clarified this sentence to read “Lack of an amino acid inference (‘?’) was considered neither an error nor a correct prediction.” (pg. 4, line 164-165)

Line 213-216 and Line 241: It is better to explain why the number of the aligned Pfam consensus columns was so small in these species.

We clarified this sentence to read:

“In three species belonging to a lineage of *Hanseniaspora* with low genomic GC content (Steenwyk et al., 2019), the arginine codons CGC and/or CGG had a ‘?’ inference due to few (<20) aligned Pfam consensus columns presumably due to rare usage of those codons.” (pg. 6, lines 218-221)

Our reasoning for the Pfam consensus columns aligned to stop codons can be found in pg. 6, lines 224-226, where we also corrected an error (“transition” instead of “transversion”).

Line 221: In the title of this paragraph, "Computational screen of all bacterial and archaeal genomes finds previously known alternative genetic codes", the authors used "all", but it is not precise. It should be removed.

Agreed. We removed the word “all”, so now the section heading reads “Computational screen of bacterial and archaeal genomes finds previously known alternative genetic codes” (pg. 8, line 227-228)

Line 238-240: Provide the reference(s) for the claim that Mycoplasmatales and Entomoplasmatales translates the opal stop codon UGA as Trp.

We added a citation to this statement: “In the Mycoplasmatales and Entomoplasmatales, which are believed to translate the canonical stop codon UGA as tryptophan (Bové, 1993), …” in pg. 8, lines 245-246.

Line 247-250: Provide the claim that Gracilibacteria translate UGA as Gly.

We added a citation to the statement: “In the Gracilibacteria, which are believed to translate the stop codon UGA as glycine (Rinke et al., 2013), …” in pg. 8, lines 253-254.

Line 282-284: Provide the reference for the argument that "high GC content-driven nonsynonymous substitutions of the AAA and AAG lysine codons to AGA and AGG arginine codons at protein residues that can tolerate either positively-charged amino acid".

High GC content-driven nonsynonymous substitutions is our proposed explanation for the cause of the AGA/AGG to lysine inference. We have clarified that this is a proposed explanation by changing the text to read

“the source of the lysine inference may come from high GC content-driven nonsynonymous substitutions of the AAA and AAG lysine codons to AGA and AGG arginine codons at protein residues that can tolerate either positively-charged amino acid.” (pg. 10, lines 288-290)

Line 287: Citing N. Sueoka (1961) that "…have long been observed to preferentially use more arginine and less lysine.". However, in his original study, Sueoka is simply plotting the possible correlation between the GC contents and the amino acid composition, drawing from a limited 'dataset' at his time. He did not directly observe any preferential usage of Arg over Lys in high-GC bacteria. This sentence should be corrected accordingly, given the citation to Sueoka's paper.

We revised the text to clarify that the relationship between genomic GC content and arginine and lysine usage is an observed correlation. The text now reads:

“Genomic GC content has long been correlated with greater arginine usage and lower lysine usage (Sueoka, 1961; Lightfield et al., 2011), possibly due to substitutions between the aforementioned lysine and arginine codons (Knight et al., 2001b).” (pg. 10, lines 292-295)

Line 300-303: The authors should be more specific in their analysis of the values of probabiltiy and tRNA sequences on which they base the results they are describing here.

We changed this sentence to remove subjective assessment of the additional evidence for the new genetic code: “We found five clades using candidate novel alternative genetic codes with additional computational evidence, such as tRNA genes consistent with the new translation.” (pg. 10, lines 305-306). Further description of the types of additional evidence examined can be found in the beginning of that section (pg. 8, lines 271-274).

Line 317-318: The values of the probability on which the author's decision were based should be indicated or summarized.

We updated this sentence to make it clear that we are describing the multiple sequence alignment shown in Figure 3B:

“In the alignment, AGG codons are used interchangeably with AUG methionine codons within the reassigned clade and tend to occur at positions conserved for methionine and other nonpolar amino acids in the other species.” (pg. 10, lines 322-324)

Line 368-376: The values of the probability on which the author's decision were based should be indicated or summarized.

We edited this sentence to remove subjective language. The sentence now reads:

“The remaining four clades with codon reassignments supported by additional computational evidence all affect the arginine codons CGA and/or CGG (Figure 4).” (pg. 13, lines 380-381)

The additional computational evidence is described in more detail in the text (pg. 13, lines 394-401) and in the figure supplements for Figure 4.

Line 408-414: Do these species in which tRNAUCG is missing have homologs of the modifying enzyme responsible for the formation of inosine?

We had the same question and have been investigating the presence of the tRNA-specific adenosine deaminase (TadA) gene in bacterial genomes. These results are beyond the scope of this paper, but a preliminary search could not find TadA in Absconditabacteria and Gracilibacteria genomes, consistent with the lack of an arginine tRNA(ACG) gene.

Line 443: The authors described "two differently charged tRNAs" here. However, they did not demonstrate that.

We revised this sentence (pg. 15-16, lines 454-456) to read:

“We validate Codetta on the well-studied reassignments of CUG in yeasts and rediscover the likely ambiguous translation of CUG as serine and leucine in some *Ascoidea* and *Saccharomycopsis* species.”

Figure 2 and line 709-715 (method section): Supply the exact sequences of T7 in vitro transcribed tRNA used as controls in northern blot.

We added the sequences of the in vitro transcription DNA templates and expected transcripts as Figure 2--Source Data 3.

Reviewer #2 (Recommendations for the authors):The paper proposes a method for large scale genome analysis that is able to detect reassignments in the genetic code. In general, the examples given are convncing. The method detects several variants that are already known and discovers some new ones. This appears to be useful development and a thorough analysis.The essentials of the method are in lines 112-126. I don't find this 100% clear. If I understand correctly, the DNA sequence of the test sequence is translated with the standard genetic code and then aligned with HMMs of proteins. If the codon follows the standard code, the amino acid will be in agreement with the common amino acid in the alignment column. If the codon has been reassigned, the apparent amino acid will be an unusual one according to the alignment profile. This requires the protein sequence to be less variable than the genetic code. I can see that this would become convincing if the same apparently wrong amino acid appears consistently in alignment columns where a particular amino acid is the most common one. How many occurrences of an apparently wrong amino acid are needed in practice to draw an inference?

This is the first point in the Essential Revisions; see above.

The only place where having an alternative translation (an “apparently wrong amino acid”) could potentially impact the ability to correctly infer a codon would be if the preliminary translation using the standard genetic code is sufficiently incorrect that it impedes the ability to detect homology with Pfam domains. We have not encountered this problem in our Codetta analyses, but this could be addressed in the future by using different preliminary translations. (We express this potential issue in pg. 3, lines 124-126.)

This paragraph says there are 17000 alignments in Pfam, but it does not say how many are used for the inference.

The number of aligned Pfam domains is depends on the number and type of protein-coding regions in the input nucleotide sequence. For a mitochondrial genome, the number of aligned Pfam domains could be in the single digits, while for a complete bacterial genome the number of Pfam domains could be in the hundreds. For the AGG->Met alternative genetic code found in a clade of Bacilli, over 480 different Pfam domains contributed to the inference of AGG (pg. 10, line 318).

There is no mention of mitochondrial genetic codes. Does this method work with mitochondrial genomes? Maybe there are too few coding sequences in mitochondria?

This is the second point in the Essential Revisions section; see above.

The case of Lys and Arg codons discussed in lines 277-291 is interesting. Since Lys and Arg are similar amino acids we expect non-synonymous substitutions to be frequent between Lys and Arg. Therefore it is not surprising that the method might give a false positive prediction of a codon reassignment. In fact it is surprising that more cases like this do not occur. For example AUA is frequently reassigned from lle to Met in mitochondria, and an Ile to Met non-synonymous substitution is also quite possible. Does this not also show up in the analysis?

We did not detect any candidate alternative genetic codes with the change of AUA Ile -> Met in our screen (the entire list of candidate alternative genetic codes can be found in Supplementary File 2). We think that the non-synonymous substitutions to AGA and AGG codons at high genomic GC content may generate a stronger “codon reassignment”-like signal than the situation at AUA codons. This is because we hypothesize there are two forces at play: (1) lysine AAA and AAG are non-synonymously substituting to the arginine codons AGA and AGG at positions that permit and (1) the arginine AGA and AGG codons are synonymously substituting to the GC-rich CGN arginine codons. Together, this may generate an even greater enrichment of lysine vs arginine conservation at AGA/AGG codons.